## Registered report

health and disease and epidemiology

infectious disease, coronavirus, 2019-nCoV, SARS-CoV-2, observational, risk of bias

**Author for correspondence:**
James A. Smith
e-mail: james.smith2@ndorms.ox.ac.uk

# Estimating the effect of COVID-19 on trial design characteristics: a registered report

James A. Smith[1,4], Nicholas DeVito[2], Hopin Lee[1], Catherine Tiplady[1], Roxanna E. Abhari[3] and Christiana Kartsonaki[3]

[1]Botnar Research Centre and Centre for Statistics in Medicine, Nuffield Department of Orthopaedics, Rheumatology and Musculoskeletal Sciences,
[2]Centre for Evidence-Based Medicine, Nuffield Department of Primary Care Health Sciences, and [3]Nuffield Department of Population Health, University of Oxford, Oxford OX1 2JD, UK
[4]National Institute for Health Research Oxford Biomedical Research Centre, John Radcliffe Hospital, Oxford OX3 9DU, UK

JAS, 0000-0003-2634-0268

There have been reports of poor-quality research during the COVID-19 pandemic. This registered report assessed design characteristics of registered clinical trials for COVID-19 compared to non-COVID-19 trials to empirically explore the design of clinical research during a pandemic and how it compares to research conducted in non-pandemic times. We did a retrospective cohort study with a 1:1 ratio of interventional COVID-19 registrations to non-COVID-19 registrations, with four trial design outcomes: use of control arm, randomization, blinding and prospective registration. Logistic regression was used to estimate the odds ratio of investigating COVID-19 versus not COVID-19 and estimate direct and total effects of investigating COVID-19 for each outcome. The primary analysis showed a positive direct and total effect of COVID-19 on the use of control arms and randomization. It showed a negative direct effect of COVID-19 on blinding but no evidence of a total effect. There was no evidence of an effect on prospective registration. Taken together with secondary and sensitivity analyses, our findings are inconclusive but point towards a higher prevalence of key design characteristics in COVID-19 trials versus controls. The findings do not support much existing COVID-19 research quality literature, which generally suggests that COVID-19 led to a reduction in quality. Limitations included some data quality issues, minor deviations from the pre-registered plan and the fact that trial registrations were analysed which may

not accurately reflect study design and conduct. Following in-principle acceptance, the approved stage 1 version of this manuscript was pre-registered on the Open Science Framework at https://doi.org/10.17605/OSF.IO/5YAEB. This pre-registration was performed prior to data analysis.

## 1. Introduction

The rush to conduct research during the COVID-19 crisis may compromise research quality and rigour. Several papers have highlighted issues with the emerging research landscape [1–7], including in clinical trials [1,8,9]. However, suboptimal trial design is common outside of pandemics (e.g. [10,11]). There is little evidence examining whether issues with research design in this pandemic are comparable to normal levels, or exceed normal levels and should be considered 'pandemic research exceptionalism' [4]. Unlike other infectious disease epidemics where relatively few studies were done [12], research efforts for COVID-19 have continued to be planned and conducted, and empirical comparisons to non-pandemic times are possible.

Clinical trial registrations contain design details that could be used to assess aspects of internal validity, or the extent to which a study is free from potential bias [13]. As of 11 November 2020, 6660 trial registrations for COVID-19 were included on the International Clinical Trials Registry Platform (ICTRP; https://www.who.int/ictrp/en/). These registrations represent a large, relatively standardized dataset for which a comparator sample can readily be developed. There are notable efforts to track and analyse COVID-19 trial registrations (for example, http://covid19.trialstracker.net/ and https://covid-evidence.org/ and [8]); however, none have collected data from a comparator sample or focused explicitly on assessing trial design.

This study aims to assess differences in key design characteristics between COVID-19 and non-COVID-19 registrations. We are primarily interested in the association between investigating COVID-19 and trial design characteristics conditional on covariates that might reasonably also impact those characteristics. There may be systematic differences between trials of COVID-19 compared to other trials because COVID-19 is a new disease, and trials investigating new diseases may differ from trials investigating established diseases: for example, they may be more likely to be early phase and therefore more likely to be uncontrolled. Using the terminology of causal modelling, we are interested in the direct effect of investigating COVID-19 on trial design, though we provide data on overall changes attributable to COVID-19, including that mediated via other variables (in causal language, the total effect). Our analyses also describe the COVID-19 trial landscape in terms of trial characteristics and assess how it has changed over time.

This work helps understand the planning and design of clinical research during a pandemic and how it compares to non-pandemic research. Such understanding is important to inform whether efforts to specifically improve conduct of research in global health emergencies are needed, or whether shortcomings represent broader problems that may require broader solutions.

## 1.1. Aims and hypotheses

Specifically, this work assessed the difference in key trial design characteristics, as measured by information contained in trial registrations, between interventional COVID-19 drug trial registrations and non-COVID-19 trial registrations sampled from prior to the start of the pandemic. We focused on interventional drug trials to improve homogeneity[1] in the samples and because there are widely accepted methods for their conduct [14]. We focused on four criteria that are considered important in interventional clinical trials: use of control arm, randomization, blinding and prospective registration.

Our primary aim was to estimate the direct effect [15] of investigating COVID-19 on prevalence of each outcome, which we investigated in our primary analysis. In secondary analysis of this direct effect, we compared COVID-19 trials and a separate control sample of trials more closely matched in indication. Many indications in the main control sample may differ substantially from the indications investigated under the umbrella of COVID-19, which may also vary.

A secondary aim was to estimate the total effect[2] [15] of investigating COVID-19 on outcomes, which we did in the main and more closely matched dataset as done for the analyses of the direct effect.

---

[1]In the stage 1 IPA this read 'heterogeneity', which was a typo.

[2]In the stage 1 IPA this read 'conditional total effect'. All effects calculated in this paper are conditional, so this word was removed to avoid confusion.

For all comparative analyses, the null hypotheses are the same: in comparison to non-COVID trials from outside the pandemic, after conditioning on relevant covariates, COVID-19 trials

1. did not differ in prevalence of use of control arms
2. did not differ in prevalence of randomization in trials with control arms
3. did not differ in prevalence of at least one form of blinding
4. did not differ in prevalence of prospective registration.

## 1.2. Other relevant studies

To identify other published studies investigating similar research questions using trial registries, we searched PubMed (supplementary material for search details) for studies published from 1 March 2020 to 27 October 2020 and Open Science Framework (OSF) for relevant project registrations (no date restrictions). Two authors (J.A.S. and C.T.) reviewed the results. We also reviewed the related research section of COVID-evidence (https://covid-evidence.org/related-research) and generally searched the Internet. No directly comparable studies were identified. We are aware of one preprint investigating a similar question based on journal articles rather than trial registrations [16].

## 1.3. Pilot study

We did a pilot study to assess the feasibility of using the ICTRP database for our sample, and to pilot other changes to our study following peer review of the stage 1 manuscript. Full details are provided on the OSF (https://osf.io/pjc9s/?view_only=1314dcb40c3640009e10fca85ba7d7aa) and the findings are referred to in the methods when appropriate.

# 2. Methods

We reported this study in accordance with strengthening the reporting of observational studies in epidemiology (STROBE) statement for cohort studies [17].

## 2.1. Study design, setting and included trials

We used a retrospective cohort design with COVID-19 trial registrations compared to historic non-COVID-19 trial registrations. In the language of a typical cohort study, investigating COVID-19 is the exposure.

The choice of control exposure (i.e. non-COVID-19 trial registrations) is important and possibly contentious. For the dataset used for our main analysis (the main dataset), we randomly sampled all non-COVID-19 drug trials, not limiting the indication being studied. We think this is the most relevant comparison because randomly sampling from all trials is a fair way to represent the typical design of trials outside the pandemic, and we wanted to assess whether the design of COVID-19 trials differs from design of trials outside of the pandemic.

In a second dataset (the indication-matched dataset), our control sample included only trials which address conditions subjectively comparable to COVID-19. This was to address possible concerns about differences in research design between types of indications (e.g. cancer trials might differ systematically from infectious disease trials) leading to observed differences in the main analysis. We selected respiratory, cardiovascular and infectious disease conditions that have symptoms or treatment approaches similar to COVID-19, then searched these conditions on clinicaltrials.gov and examined search details for relevant synonyms or similar conditions (selected conditions listed in inclusion criteria). Determining what is a comparable indication is inherently subjective: for example, non-COVID-19 pneumonia seems an obvious comparator; however, non-COVID-19 pneumonia was the leading cause of death for children under 5 in 2017 [18], whereas the risk to children of COVID-19 is thought to be extremely low [19], so in some senses, they are clearly not comparable. Because of this subjectivity, analyses using this dataset were secondary and included to check the robustness of our findings to changes in the control sample.

Therefore, we developed two datasets: the main dataset and the indication-matched dataset. Both datasets had a COVID-19 : non-COVID-19 ratio of 1 : 1 and were derived from the ICTRP. ICTRP provides access to a database of trials from 18 trial registries internationally and is the most

comprehensive database of global trials available. COVID-19 trials were identified from the ICTRP export of COVID-19 trials (https://www.who.int/ictrp/en/) when we began this study. The control sample for the main dataset was identified from the full ICTRP data download (https://www.who.int/clinical-trials-registry-platform) and included trials with a registration date from 1 January 2019 to the same date in 2019 that we download the COVID-19 sample in 2020. For the indication-matched dataset, we used the same COVID-19 trials as the main dataset, and the control arm was identified from the ICTRP full data export. The control arm was from 2016 to 2019[3] to ensure that a sufficient sample size is available. To select trials for our datasets, we applied relevant filters to the data exports (date restrictions, condition searches and interventional trial filter) and then randomly ordered the resulting datasets. Registry entries were then reviewed in that random order and trials included and excluded until the desired sample size was met.

We used recent historical, rather than contemporaneous, control samples for our datasets, because it is likely that there was an effect of the pandemic on planning of non-COVID-19 studies. Supporting this, new clinical trial starts decreased from January to May 2020 [20], and many trials were stopped due to COVID-19 [21–23]. We thought it was unlikely that trial characteristics would have changed meaningfully due to the passage of 1–2 years.

## 2.2. Inclusion and exclusion criteria

### 2.2.1. All datasets

Inclusion criteria

— Interventional trials, as determined by the study type specified in the ICTRP data export
— At least one arm of the trial investigating a drug (including small molecules, antibodies, proteins, blood-derived products (e.g. plasma), biologicals, ATMPs, vaccines, herbal therapies and vitamin therapies)[4]
— For the COVID-19 arm:
  o Investigating a condition including the term 'COVID-19' (or any of the synonyms listed in supplementary material) in the listed 'condition' in the ICTRP export
  o First posted from 1 January 2020

Exclusion criteria:

— Trials with status of withdrawn, defined as 'Study halted prematurely, prior to enrolment of first participant' [24].
— Studies where the intervention is combined (aka drugs given alongside another intervention)[5]

---

[3]Deviation: this data range was expanded during data collection with approval from the editor. This range was chosen to meet the required sample size for the indication matched comparison after false positive terms were excluded.

[4]Post stage 1 acceptance clarification: It was not always clear from information in the trial registry whether or not alternative medicines were drugs (e.g. herbal therapies), particularly because the authors screening the trials were from western medicine backgrounds. This was particularly the case for medicines described as traditional Chinese, siddha, unani or ayurveda medicines/therapies. If we were unable to determine from the trial registry what the intervention was, we excluded the trial on the basis that we did not know that it was a drug: e.g. if the intervention was just described as 'siddha therapy'. If more details were given, we searched the internet to determine whether or not the intervention should be considered a drug, and only included it when we could identify the constituents of the intervention and verify they were drugs. Often, this meant determining whether or not it was a herbal therapy, and as well as generally searching the internet we relied on two online lists of herbal therapies (at the time of writing available http://www.indiahomeclub.com/botanical_garden/medicinal_plants_uesd_in_ays.html and https://en.wikipedia.org/wiki/Chinese_herbology#50_fundamental_herbs). If there were multiple components to the intervention, we checked that they were all drugs before considering the intervention to be a drug. There were some interventions that we had to discuss the inclusion of, to determine whether or not they met our criteria. We decided to include the following: (i) Fecal transplants; (ii) Metals such as magnesium and iron; (iii) Plasmapheresis (as a blood-derived product); (iv) E-cigarettes, patches and gum. We decided to exclude the following as not drugs: (i) Ozone therapy and similar therapies such as ozonated autohemotherapy; (ii) Oxygen therapy; (iii) Hydrogen gas; (iv) Foodstuffs; (v) Aromatherapy; (vi) Beeswax or similar animal products; (vii) Green tea, unless details were given of the ingredients of the tea and they could be determined to be drugs according to other criteria; (viii) Dietary supplements (including amino acids), unless they met other criteria for being a drug (e.g. they were a vitamin); (ix) Diagnostic tests, including cases when a drug was administered purely for diagnostic reasons; (x) Fluid therapy, unless a specific drug was given.

[5]Post stage 1 acceptance clarification: This statement was intended to exclude e.g. drug-device combination products in comparison to e.g. placebo, since we would not be able to say that the study is testing the drug specifically). However, if interpreted strictly, many studies would be excluded because, for example, participants were given the drug alongside standard treatment, which are often not limited only to drug treatment (whether specified or not). If a drug with standard treatment was tested against placebo with standard treatment alone or placebo plus standard treatment, we felt this should not be excluded on the basis of being a combined

— Homeopathic treatments
— For control arms:
  o Trials whose recruitment criteria have been adapted to include COVID-19 patients
— For trials registered on EU Clinical Trials Register (EUCTR), the same trial conducted in different countries is often represented by several different trial registration numbers. Duplicate EUCTR trials were excluded on the basis of the trial URL[6].

## 2.2.2. Main dataset control arm

Inclusion criteria:

— Investigating any condition other than COVID-19 as defined above
— Registration date from 1 January 2019 to the same date in 2019 that we download the COVID-19 sample in 2020

## 2.2.3. Indication-matched dataset control arm

Inclusion criteria:

— Investigating a condition including one of the following terms in the ICTRP export but not COVID-19 as defined above:
  o septic shock
  o multi-organ failure; multiple organ failure, multiple organ dysfunction syndrome, multiple systems organ failure, multi-system organ failure
  o cardiogenic shock
  o myocarditis, myocardial inflammation
  o myocardial ischaemia
  o respiratory failure, respiratory insufficiency
  o ARDS, respiratory distress syndrome
  o Pneumonia
  o influenza, flu
  o respiratory arrest, apnea, breathing cessation, breathing stops, pulmonary arrest
  o cardiac arrest, heart attack, asystole, asystolia, asystolic
— Registration date from 1 January 2016 to 31 December 2019.

Exclusion criteria[7]:

— Because the term 'Flu' returns false positives, we removed trials with conditions including the following terms: 'fluor', 'influenc', 'fluid', 'flush', 'effluvium', 'reflux', 'flutter', 'flurane', 'Flurpiridaz', 'fluent', 'leflunomide', 'Fluconazole', 'fludarabine', 'fluctuat', 'fluoxetine', 'Scheimpflug', 'TRIFLURIDINE', 'flucytosine' and 'fluoxetine'
— Because the term 'ards' returns false positives, we removed trials including these terms in the conditions: 'Stewards', 'hazards', 'towards', 'wards', 'regards', 'retardSystem' and 'Richardson'

---

intervention, since the drug is the variable that changes between the arms and is therefore the intervention being investigated. We therefore included such trials. We extended this reasoning to drugs combined with procedures, radiation, delivery devices, or similar: if a drug was given during a procedure (e.g. infusion), for example, and the variable between at least two of the arms was the drug rather than the procedure, we included the trial. If, however, there was a trial where a drug was given during a procedure in one arm, and a different drug was given during a different procedure in another arm, we excluded the study on the basis that it was not only the drug being tested, but rather the procedure : drug combination. If there was at least one arm of the trial where only the drug differed compared to another arm, then we included the trial. If it was a single arm trial, we made a judgement about whether or not it was the drug that was being studied in the trial.

[6]Deviation: This criterion was added during data collection with approval of the editor.

[7]Deviation: These criteria were added during data collection, before any data analysis, with approval of the editor.

**Table 1.** Details of outcomes used as measures of risk of bias in trial design.

| outcome | definition | options | justification |
|---|---|---|---|
| use of control arm | Participants were assigned to one of at least two study arms. Historical controls were not considered controls. | Yes/No[a] | For assessing interventions, the use of a control arm is widely considered to be essential when it is feasible. |
| randomization | Patients are randomized to one of at least two study arms, by any method. If not stated or single group we considered the study non-randomized. | Yes/No[a] | Randomization is considered the ideal approach to clinical studies when it is possible. Problems in randomization are associated with differences in intervention effect estimates [25,26]. |
| blinding (masking) | At least one of subjects, research investigators, outcome assessors or data analysts are blinded (masked) to information about the intervention. If not stated, we considered the study not blinded. | Yes/No[a] | Blinding, in some form (e.g. of outcome assessment), is part of existing risk of bias tools [27,28] and is applicable to any study type. We also collected data to be presented descriptively on who was blinded, where this information is available. |
| prospective registration | The trial was registered on or before the date on which it was intended to be commenced (see elaboration in text). | Yes/No[a] | Prospective registration is considered best practice and required by ICMJE [29]. Lack of prospective registration may be associated with larger treatment effect sizes [30]. |

[a]if not stated, we assumed 'No'. In all cases, this assumption was recorded.

## 2.3. Variables

### 2.3.1. Outcome variables

Outcome variables were the use of control arm, randomization, blinding and prospective registration. Definitions and justifications are given in table 1. These variables were chosen because they are often used in assessments of risk of bias of trials publications [13,27,28], and because we think they give a reasonable measure of the overall quality of the trials being conducted. In choosing outcomes, we also considered the practicality of retrieving accurate information and made common sense judgements. Each outcome was dichotomous for the main analyses.

We followed previous literature in classifying trials without information on outcomes: trials with a single group were considered non-randomized and unblinded [31,32], trials with more than one group were considered to have a control arm, and trials were considered unblinded if it was not reported and could not be inferred from other information in the registry record [32]. Trials that were not stated as being randomized (single or multi-arm) were considered non-randomized. When this information was inferred rather than stated explicitly, this was recorded.

For prospective registration, we recorded whether the trial registration date was prior to the start date on the source registry. We consider that the important outcome is whether the trial has been registered prospectively on at least one registry (acknowledging that it is still best practice to prospectively register a trial wherever it is registered). ICTRP data includes a 'bridging flag' that indicates whether a trial has been registered on a registry other than the one for which data is provided. We manually reviewed all trials with a bridging flag to identify the other trial registrations and considered a trial prospectively registered if it was prospectively registered in at least one registry. If there was no statement regarding prospective registration and no start date or registration date was found in all registry entries, we considered the study retrospectively registered.

There are limitations in treating blinding dichotomously as an outcome. For example, the importance of blinding may depend on the subjectivity of the outcome and who exactly is blinded. There is also uncertainty about the importance of blinding in trials [33]. We included this outcome because we felt that most assessments of trial quality would include it, but it should be interpreted in the context of these limitations. Where possible we collected information on who was blinded in the trial and reported these data. For the main analysis, we treated blinding as dichotomous because we expected details of blinding to be missing in some cases (pilot study and [34,35]).

In our analyses, rather than combining outcomes into a single score, we analysed each item separately, which is a recommended approach when appraising trial results [13]. There is varying acceptance of the importance of each item, in particular blinding, and we leave judgement of relative weighting to the reader.

### 2.3.2. Other variables

We collected data on other variables to adjust for during analysis (electronic supplementary material, table S9 for full details). The following variables were collected for both datasets: source registry, phase of trial, geographical region (or regions) of trial location, whether the trial was single or multi-centre, intervention type, the primary purpose of the trial, sponsor type, sample size of the trial and type (or types) of intervention. We also collected the start date of the trial for descriptive purposes.

## 2.4. Data sources/measurement

Where possible, we automated the extraction of the variables from the data exports from ICTRP or source registry data exports using R [36] and Python [37]. To ensure the accuracy of automated extraction and identify data that require manual extraction, two authors wrote code to extract the data (one in R and one in Python [some relevant Python code already had been written in another project [38]]) and compared results across datasets. Discrepancies in results were examined, and if they were attributable to errors in code, the code was corrected, analysis re-run and results again compared. This process was repeated until analyses converged (see also outcome neutral criterion 1). Any discrepancies that could not be reliably addressed through automation were reviewed and manually extracted by two authors.

Manual data extraction was done in Microsoft Excel. Data validation tools in Excel were used to reduce the probability of errors. When manual extraction was relied on, data was extracted in duplicate by two independent reviewers, extracted datasets were compared in R and discrepancies were consolidated through discussion.

## 2.5. Limiting bias

In our registered protocol, we provided the code that was used for randomly ordering trials, including the seed, and specified the analysis code we used for our analyses (electronic supplementary material, analysis document). Blinding of data extraction to group allocation was not feasible, because for some items we needed to consult the trial title or full trial registration, which contains information about the indication.

## 2.6. Study size

We conducted a power analysis to determine the required sample size for our primary analysis (code provided in the GitHub repository). The same power calculation was applicable to the other analyses. We aimed to detect a minimum difference of 10% prevalence on each of the four outcomes using an overall alpha of 0.05 and power of 0.95, assuming a 1 : 1 ratio of COVID-19:control registrations. Ten per cent was an arbitrary choice. We did not know the true probability of the outcome in either group, so we made the conservative assumption of 0.5 in one group. The total sample size required to detect a 10% difference in prevalence (i.e. to 0.6 or 0.4) was 1694 (847 per arm), using a Bonferroni correction to account for four comparisons.

There were ample trials available for the COVID-19 trials and main dataset control arm. To check if there were sufficient trials for the indication-matched control arm, we searched clinicaltrials.gov for the indications in our inclusion criteria and found that in 2018 and 2019, there were 1120 trials that would likely be applicable. Since clinicaltrials.gov only represents about 36% of trials in ICTRP over that time period (author's calculations), we expected there to be sufficient trials available. As explained above,

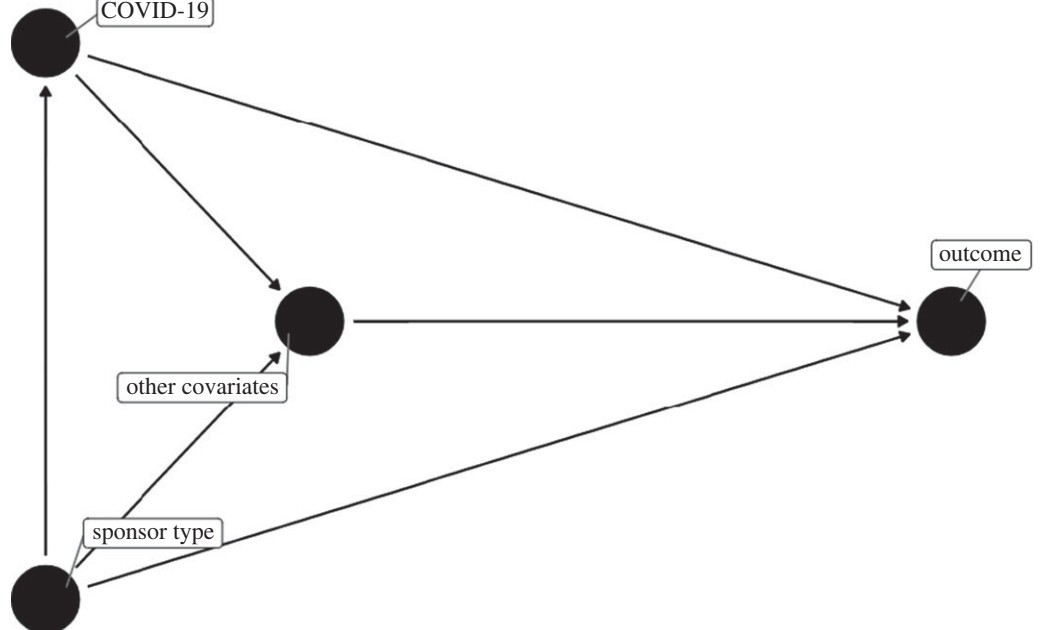

**Figure 1.** DAG [39] showing assumptions regarding relationships between variables in our study. Nodes represent variables for which we collected data, and edges represent the assumed causal pathways between variables. All mediator variables have been included in a single node for clarity of presentation (other covariates). Sponsor type is a confounder. 'COVID-19' means studying COVID-19 as the indication in a trial and is the exposure. 'Outcome' is one of the use of control arm, randomization, blinding or prospective registration; 'Other covariates' are source registry, phase, geographical region(s), multi- versus single-centre, primary purpose, sample size and intervention type(s). We have not included some relationships between 'Other covariates' because they do not impact interpretation of the DAG. For example, phase of the trial likely influences whether the trial is multi- versus single-centre.

we ultimately had to expand this date range after realizing that many false positives were included in the original searches.

## 2.7. Quantitative variables

Primary outcomes were all treated as binary variables. Continuous variables were kept continuous in analysis, categorical variables were treated as categorical and binary variables analysed as binary. For the continuous covariate (sample size), we examined linearity in the logit by splitting the covariate into quintile groups, calculating the log odds for each group and plotting the median of each group against the log odds. The relationship was not perfectly linear, so we log–transformed the covariate, repeated the process and observed an improvement. The natural logarithm of the sample size was therefore used for all modelling.

## 2.8. Statistical methods

### 2.8.1. Relationship between collected variables

We generated a directed acyclic graph (DAG) to display our understanding and assumptions about the relationship between collected variables and aid interpretation of our analyses [15] (figure 1). It is plausible that investigating COVID-19 in a trial could influence many covariates, and that these could in turn influence the outcome. We thought that covariates or the outcome influencing the investigation of COVID-19 were less plausible, with the exception that characteristics of the sponsor could influence both the decision to study COVID-19 and the outcome. We did not anticipate interactions between investigating COVID-19 and covariates. We considered the same relationships between variables to be applicable to all four outcomes. According to this representation, all covariates apart from sponsor type were mediators of the effect of COVID-19 and only sponsor type was a confounder (defined as a variable that affects both the exposure and outcome [40]).

According to this DAG, adjustment for all collected covariates in a regression model allows estimation of the direct effect of investigating COVID-19 on each outcome, i.e. the effect of COVID-19 on outcome that is not mediated via the effect of COVID-19 on covariates. This interpretation requires that there is no unmeasured confounding between (i) COVID-19 and outcome (ii) COVID-19 and other covariates and (iii) any other covariates and outcome [15]. Adjusting only for sponsor type closes the only open path to outcome and allow estimation of the total effect of COVID-19 on outcome, i.e. the effect of COVID-19 on outcome, including that mediated by the effect of COVID-19 on other covariates. This interpretation requires that there is no unmeasured confounding of the COVID-19 : outcome relationship.

The relationship between geographical region and COVID-19 warrants further discussion. Geographical regions are variables representing trial locations. In the DAG, we have included geographical regions as mediator variables: it can be argued that investigators decide to study COVID-19 and subsequently choose suitable trial locations to do that study. However, sponsors could also have access to particular trial locations, and those locations could influence the decision to study COVID-19. In that case, location could be considered a confounder rather than a mediator. Both arguments are plausible, so we assessed the impact of the decision to treat regions as mediators in sensitivity analysis of the total effect (for the direct effect, we adjusted for mediators and confounders so there was no impact).

### 2.8.1.1. Sources of residual confounding

After adjustment for sponsor type, it is likely that there will be residual confounding. Factors such as local research culture, availability of resources, and the funding source could plausibly confound both the COVID-19 : outcome and the other covariate : outcome relationships. For example, it seems likely that the specific investigators on a trial could influence the decision to study COVID-19, phase of the trial and all of the outcomes. Similarly, funders may have calls specifically for COVID-19 trials, require that all funded trials have certain design characteristics (e.g. blinding) and limit funding to trials conducted in certain geographical regions, which might in turn affect the trial design. Unfortunately, in clinical trial registrations, information that would allow us to account for additional sources of confounding are unavailable.

### 2.8.1.2. Model interpretation

The use of the terms direct or total effect of investigating COVID-19 in this paper should therefore be interpreted with the understanding that they refer to the scenario in which our DAG is correct and complete, and there is no unmeasured confounding. More strictly, when using the term direct effect, we are referring to the association of a trial investigating COVID-19 with the probability of the outcome, conditional on all measured covariates. When using the term total effect, we are referring to the association of a trial investigating COVID-19 with the probability of the outcome, conditional on the measured covariate we have identified as a confounder: sponsor type (and geographical regions in sensitivity analysis).

### 2.8.2. Analyses

All analyses are numbered for clarity when reporting the results. To illustrate our analyses, in the supplementary analysis document accompanying the pre-registered protocol, we included a simulated dataset with the variables we expected our final dataset to have. The code we intended to use for the logistic regression models was provided.

### 2.8.2.1. Data pre-processing

Missing data in covariates was possible, though outcomes were defined such that there should be no missing data. We used multiple imputation by chained equations to impute missing values in covariates [41].

In terms of outlier identification, sample size was the only continuous variable included in the datasets and we plotted a histogram (not presented in the results; see §1 here: https://github.com/worcjamessmith/COVID-trial-characteristics/blob/main/Analysis-code.html) to identify implausible values.

If there were problems with convergence or predicted probabilities of 0 or 1, we removed variables from the regression model to diagnose which were causing the problem, and then grouped infrequent categories until the model worked properly. The trial registry variable was problematic because there were many categories with very few entries. We therefore kept the top five registries in terms of frequency across all three arms of the dataset as separate categories and grouped the remaining registries into a single 'other' category[8].

### 2.8.2.2. Descriptive analysis

For each dataset, we presented unadjusted prevalence of all characteristics for which data were collected, summarized by group. Binary and categorical variables were presented as count ($N$) and percentage[9]. Sample size (the only continuous variable) was presented as median ± interquartile range. We also included a flow chart illustrating the flow of trial registrations through the study.

### 2.8.2.3. COVID-19 outcomes over time

It was plausible that quality of COVID-19 registrations could have improved or otherwise changed over time. We examined this by plotting and presenting the prevalence of main outcomes for the trials in the COVID-19 arm of the main dataset over time, grouped by month according to start date. This was interpreted qualitatively.

### 2.8.3. Regression analyses

For each regression analysis (analyses 4–7), we estimated the odds ratio of investigating COVID-19 on each outcome using logistic regression. We calculated 95% CIs for the odds ratio and test for a difference of the odds ratio for COVID-19 from one using a Wald test (two-sided) with an alpha across all tests in each analysis of 0.05. For each analysis, we reported a Bonferroni corrected (four comparisons) 95% CI and $p$-value for each outcome. Full model coefficients are presented in the project's GitHub repository[10] to avoid misinterpretation [15]. Analyses and pre-specified interpretation are summarized in table 2.

### 2.8.4. Primary analysis

### 2.8.4.1. Main dataset: direct effect

Our primary analysis presented the association (i.e. odds ratio) of investigating COVID-19 on each outcome in the main dataset, conditional on all measured covariates: i.e. the direct effect of COVID-19, given the interpretation discussed above.

Specifically, for each of the four outcomes the model was[11] as follows:

$$
\begin{aligned}
\text{Logit ( outcome} \mid \text{covariates)} = \beta_0 + {}& \beta_1 \text{COVID} + \beta_2 \text{Source registry} + \beta_3 \text{Phase} + \beta_4 \text{Region} \\
& + \beta_5 \text{Multicentre} + \beta_6 \text{Intervention type} + \beta_7 \text{Primary purpose} \quad (2.1) \\
& + \beta_8 \text{Sponsor type} + \beta_9 \ln(\text{Sample size})
\end{aligned}
$$

where each variable was of the type and values specified in the supplementary data dictionary.

---

[8]Post stage 1 clarification: on 19 May 2021, we confirmed that this approach was acceptable with the editor. It was not specified what we would do in a scenario of problems with convergence in the original report. This decision was made prior to running analyses on the full dataset.

[9]The stage 1 article stated that we would report proportion and standard deviation of binary and categorical variables. However, this was an error: count and % is more commonly reported and more informative and was instead reported. This decision has no impact on any inferences from the study and was not communicated to the editor prior to submission of the stage 2 article. s.d. can be calculated from $N$ and percentage.

[10]In the stage 1 IPA, this read 'in the electronic supplementary material'. Because so many models were developed, this information is instead stored in the GitHub repository to prevent the electronic supplementary material becoming overly long and complex.

[11]Correction: Following stage 1 acceptance, a correction was made to this equation with permission from the editor to remove a duplicated 'intervention type' variable. The equation was also updated to reflect the fact that sample size was log-transformed.

**Table 2.** Summary of research question, hypothesis, sampling plan, outcome type, analysis plan and interpretation (LR = logistic regression).

| question | hypothesis | | sampling plan | analysis plan (# in methods) | interpretation given different outcomes |
|---|---|---|---|---|---|
| primary aim: direct effect (given figure 1 and assumptions discussed in text) | | | | | |
| 1. What is the association between investigating COVID-19 and use of control arms? | odds ratio ≠ | 1 | Main dataset: all COVID-19 trials meeting our inclusion/ exclusion criteria, matched 1 : 1 with a random sample of non-COVID-19 trials. | LR of main dataset adjusted for all covariates (4) and Wald test (two-sided) of coefficient, with *p*-value and CI adjusted for four comparisons. | To conclude that there was an association between COVID-19 and the outcome[13], we required that the *p*-value for the analysis of the main dataset was less than or equal to 0.05, and that the confidence interval for the direct effect of investigating COVID-19 in the indication-matched dataset (analysis 5) was consistent with the direction of effect observed in the primary analysis, i.e. that at least part of the confidence interval was in the direction of the effect in the primary analysis, even if the point estimate was not. If an effect was found in the primary analysis but analysis of the indication-matched dataset contradicted it, we concluded that there may be an association between COVID-19 and the outcome, but that the relationship was not robust to the change in the control sample. If the confidence interval for the COVID-19 odds ratio in the main dataset overlapped with 1, we concluded that we did not find evidence of a difference in the outcome prevalence attributable to COVID-19. *p*-values were interpreted according to table 3. |
| | | | Indication-matched dataset: the same COVID-19 trials as the main dataset, and a 1 : 1 matched control arm including trials with the conditions listed in the inclusion criteria. | LR of indication-matched dataset adjusted for all covariates (5) and Wald test of coefficient, with *p*-value and CI adjusted for four comparisons. | |

*(Continued.)*

**Table 2.** (Continued.)

| question | hypothesis | sampling plan | analysis plan (# in methods) | interpretation given different outcomes |
|---|---|---|---|---|
| 2. What is the association between investigating COVID-19 and prevalence of randomization? | odds ratio ≠ 1 | As per question 1. | As per question 1. | As per question 1. |
| 3. What is the association between investigating COVID-19 and prevalence of at least one form of blinding? | odds ratio ≠ 1 | As per question 1. | As per question 1. | As per question 1. |
| 4. What is the association between investigating COVID-19 and prevalence of prospective registration? | odds ratio ≠ 1 | As per question 1. | As per question 1. | As per question 1. |
| secondary aim: total effect (given figure 1 and assumptions discussed in text) | | | | |
| 5. What is the association between investigating COVID-19 and use of control arms? | odds ratio ≠ 1 | As per question 1. | LR of main dataset adjusted for the confounder (6) and Wald test of coefficient, with *p*-value and CI adjusted for four comparisons. LR of indication-matched dataset adjusted for the confounder (7) and Wald test of coefficient, with *p*-value and CI adjusted for four comparisons. | As per question 1. |

| | | | | |
|---|---|---|---|---|
| 6. What is the association between investigating COVID-19 and prevalence of randomization? | odds ratio ≠ 1 | As per question 1. | As per question 5. | As per question 1. |
| 7. What is the association between investigating COVID-19 and prevalence of at least one form of blinding? | odds ratio ≠ 1 | As per question 1. | As per question 5. | As per question 1. |
| 8. What is the association between investigating COVID-19 and prevalence of prospective registration? | odds ratio ≠ 1 | As per question 1. | As per question 5. | As per question 1. |

### 2.8.5. Secondary analyses

#### 2.8.5.1. Indication-matched dataset: direct effect

We planned to repeat analysis 4 (Main dataset: direct effect) for the indication-matched dataset (equation (2.1)). However, there were problems with convergence attributable to the intervention type variable, and this was therefore dropped from the models.

#### 2.8.5.2. Main dataset: total effect

We presented the association (i.e. odds ratio) of investigating COVID-19 on each outcome in the main dataset adjusting only for sponsor type, which was identified as a potential confounder: i.e. the total effect of COVID-19, given the interpretation discussed above.

Specifically, for each of the four outcomes, the model was as follows:

$$\text{Logit ( outcome | covariates)} = \beta_0 + \beta_1 \text{ COVID} + \beta_2 \text{Sponsor type.} \tag{2.2}$$

#### 2.8.5.3. Indication-matched dataset: total effect

Analysis 6 (Main dataset: total effect) was repeated for the indication-matched dataset (equation (2.2)).

### 2.8.6. Sensitivity analysis

#### 2.8.6.1. *E*-values

If the confidence intervals of the odds ratios for COVID-19 in the analyses of the main dataset (analysis 4 and 6) did not overlap with one, we calculated *e*-values [42,43] for the odds ratios and confidence intervals closest to one. The calculation of *e*-values for odds ratios relies on an approximate conversion of odds ratios to relative risk (specifically, the relative risk is calculated as the square root of the odds ratio [44]). We provided the calculated relative risk and the *e*-value for the point estimate and confidence interval closest to the null.

#### 2.8.6.2. Geographical regions as confounders

We repeated the total effect analyses (analyses 6 and 7), adjusting for geographical region as well as sponsor type, given that there was some uncertainty in the relationship between those variables and investigating COVID-19.

Specifically, for each of the four outcomes the model was as follows:

$$\text{Logit ( outcome | covariates)} = \beta_0 + \beta_1 \text{ COVID} + \beta_2 \text{Sponsor type} + \beta_3 \text{Region} + \varepsilon. \tag{2.3}$$

#### 2.8.6.3. Analysis without assumed outcomes

We repeated analyses 4–7 using only those observations for which we did not infer outcome data. This was to check whether decisions to infer outcomes might have differentially affected each group and therefore biased the findings (e.g. if COVID-19 trials tend to include less information as they are more recently registered). To avoid repeating the multiple imputation procedure several times, which is computationally expensive, and given that there was very little missing data, this sensitivity analysis was done with complete cases only[12]. This analysis was not done for prospective registration as no outcomes were assumed.

## 2.9. Interpreting results

For the analyses of direct effects, in the primary analysis (analysis 4), for each outcome, we interpreted *p*-values according to the corrected scale presented in table 3. To conclude that there was a difference in the observed proportion for a specific outcome, we required that the *p*-value for the analysis of the main dataset was less than or equal to 0.05, and that the confidence interval for the direct effect of investigating COVID-19 in the indication-matched dataset (analysis 5) was consistent with the direction of effect

---

[12]This sentence was added after stage one acceptance and was not agreed with the editor prior to submission of the stage 2 article. We think that this risk of bias introduced by this decision is minimal.

**Table 3.** Strength of evidence against the null and *p*-values used in this study. Corrected *p*-values are Bonferroni corrected for four comparisons. The evidence level refers to the level of evidence that we consider the *p*-value to provide against the specific null hypothesis being tested (Table 2).

| evidence level | *p*-value (no correction) | corrected *p*-value |
|---|---|---|
| weak | $0.01 < p \leq 0.05$ | $0.0025 < p \leq 0.0125$ |
| moderate | $0.001 < p \leq 0.01$ | $0.00025 < p \leq 0.0025$ |
| strong | $0.0001 < p \leq 0.001$ | $0.000025 < p \leq 0.00025$ |
| very strong | $p < 0.0001$ | $p < 0.000025$ |

observed in the primary analysis, i.e. that at least part of the confidence interval was in the direction of the effect in the primary analysis, even if the point estimate was not. If an effect was found in the primary analysis but analysis of the indication-matched dataset contradicted it, we concluded that there may be an association between COVID-19 and the outcome, but that the relationship was not robust to the change in the control sample. If the confidence interval for the COVID-19 odds ratio in the main dataset overlapped with 1, we concluded that we did not find evidence of a difference in the outcome prevalence attributable to COVID-19. We reported whether or not interpretation changed as a result of sensitivity analysis (analysis 10).

We interpreted the findings of analyses of total effects (analyses 6 and 7) in the same way. We reported whether or not the interpretation changed as a result of the sensitivity analyses (analysis 9 and 10).

The *e*-value (analysis 8) was interpreted as recommended [42]: as the strength of a confounder that is associated with both the treatment and outcome, above and beyond the measured confounders, on the risk ratio scale, that could move the point estimate or confidence interval of the effect to one, from the approximated risk ratio used in the calculation.

## 2.10. Summary table

### 2.10.1. Outcome neutral criteria and quality control

#### 2.10.1.1. Ninety-five per cent column-wise agreement per registry between manual and automated data extraction, where automated data extraction is relied on

We checked for errors in automated extraction by comparison to manually extracted data. We expected that the ability to automate data extraction would depend on source registry and the specific variable and its presentation in the ICTRP export or source registry export. One author manually extracted data on a random sample of 15% of each arm of the datasets (i.e. three arms in total), stratified by source registry, blind to the results of automated extraction. Manually extracted data was compared to automated data extraction in that sample and discrepancies examined. Code was improved, if possible, if there were problems with automatically extracted data. If discrepancies were due to errors in manual extraction they were corrected. To rely on automated extraction for a variable within a registry, we required that there was ultimately greater than 95% agreement between the automated and manual extraction for that variable and registry.

#### 2.10.1.2. Hundred per cent agreement between automated extractions

We required that the two efforts to automate data extraction converged.

#### 2.10.1.3. Identify inconsistencies in outcome information in trial registrations

We checked whether the information contained in fields for the primary outcomes accurately reflected other information in trial registrations. To automate data analysis, we relied on data fields in ICTRP or source registries accurately reflecting what is planned in the trial. However, investigators might input this information incorrectly, and there may be contradictory information in trial registrations. For example, the study title might state that it is a randomized controlled trial, whereas the study design field might state that group allocation was not randomized. In the 15% of trials with manual data extraction, we therefore also extracted data on whether we found contradictory information in

the trial registration for each outcome. Percentage of registrations with contradictory information was calculated per outcome. If there was greater than 10% contradictory information for any outcome, we tried to determine which information sources were accurate and corrected the data.

#### 2.10.1.4. Check of random data sample prior to submission

Prior to submission of the stage 2 report and after completion of all analysis to be used in the report, we checked a random sample of 30 trial registrations from each arm of the datasets (three in total) to ensure that there had not been any major data management issues resulting in e.g. incorrect group allocation, shuffled columns or rows, or consistently incorrect data entries.

### 2.11. Extent of prior data observation

Before submitting the version of the stage 1 registered report that was accepted, we did a pilot study, extracting data on 50 non-COVID-19 and 50 COVID-19 trials which could overlap with the sample we used for our study: to identify COVID-19 trials the COVID-19 csv export from ICTRP was downloaded, clinicaltrials.gov trials were removed, and the results randomly sampled. Full details of the pilot and the data analyses are provided on OSF (https://osf.io/pjc9s/?view_only= 1314dcb40c3640009e10fca85ba7d7aa). For the control arm, the full ICTRP export was downloaded, filtered by date, clinicaltrials.gov registrations were removed, and the results were randomly sampled. This dataset was used to assess the proportion of studies registered in ICTRP from clinicaltrials.gov. We downloaded the csv export of COVID-19 trial registrations from clinicaltrials.gov and filtered the dataset to assess the minimum number of interventional studies that would be available for the initial submission of this study. To identify relevant conditions to compare to COVID-19, we generated a pivot table in Microsoft Excel to see which conditions were most commonly studied under the heading COVID-19. Searches of clinicaltrials.gov were done to determine the number of potentially eligible interventional trials for the indication-matched dataset. Two authors (J.A.S. and R.E.A.) collected data for a study of mesenchymal stem cell trial registrations, of which seven studies were posted in clinicaltrials.gov and could overlap with the included studies in the COVID-19 arm of this study [45]. One author (N.D.) collates data for the previously mentioned COVID-19 Trials Tracker and has described subsets of COVID-19 trial designs based on specific treatments (corticosteroids, interferons, hydroxychloroquine/ chloroquine and remdesivir).

# 3. Results

## 3.1. Outcome neutral criteria and quality control

### 3.1.1. Ninety-five per cent column-wise agreement per registry between manual and automated data extraction, where automated data-extraction is relied on

Agreement between automated and manual extraction was generally good. Greater than 95% agreement was achieved for all variables used in analysis after corrections were made to the automatic and manually extracted data, except for prospective registration and start date. The comparison with prospective registration highlighted that the ICTRP date for trial registration date for clinicaltrials.gov trials was the date the trial was first submitted to clinicaltrials.gov, not the first posted date as we had expected and as was used for the manual data extraction. The use of either date is defensible and was not specified in our protocol; we decided to use the date provided to ICTRP for the main analysis, but we did not update the manual extraction to reflect this so that the difference between the approaches was apparent. Ten of 182 clinicaltrials.gov trials in the comparison (5.5%) had different values for prospective registration when using date posted versus date first submitted.

For start date, there was one discrepancy between dates from the Pan African Clinical Trials Registry (PACTR) out of 13 trials (92.3% agreement), which highlighted that when the actual start date, as opposed to target start date, was available from PACTR it was not included in the ICTRP data. We therefore manually reviewed all PACTR start dates and corrected them where needed. We did not compare blinding of specific parties, as this variable was extracted only for descriptive purposes.

### 3.1.2. Hundred per cent agreement between automated extractions

Before comparison between automated and manual data extraction efforts (outcome neutral criterion 1), automated extraction efforts converged.

### 3.1.3. Identify inconsistencies in outcome information in trial registrations

No outcome had greater than 10% contradictory information. The number (%) of contradictions recorded out of 372 included and manually extracted trials for control arm, randomization, blinding and prospective registration variables was 0 (0%), 3 (0.8%), 8 (2.2%) and 11 (3.0%), respectively.

### 3.1.4. Check of random data sample prior to submission

The check was completed on 19 July 2022 by J.A.S. No issues were identified.

## 3.2. Analyses

### 3.2.1. Data pre-processing

There was a small amount of missing data (table 4). The only variables with missing data used in analysis at the end of data pre-processing were phase ($N = 43$), sponsor type ($N = 61$) and sample size ($N = 4$). Multiple imputation by chained equations was therefore used using the R package 'mice'. Values for these three variables were imputed using all variables. Five iterations were used and 10 imputed datasets generated. Phase and sponsor type were imputed using multinomial logistic regression and sample size was imputed using predictive mean matching. Forty iterations were done to plot and inspect imputation trace lines to check that there are no trends or convergence issues. The densities of the imputed variables were visually inspected. The analyses were done on the imputed datasets and estimates were pooled using Rubin's rules [46].

A histogram of sample size did not reveal any erroneous outliers.

### 3.2.2. Descriptive analysis

Figure 2 summarizes the flow of trial registrations through the study. Descriptive summaries of the data are provided in table 4. Eight hundred and forty-seven trial registrations were initially included in each arm of the dataset; however, some were excluded during data extraction when it was observed that they did not meet inclusion criteria (e.g. they were withdrawn). This was the case more frequently in the COVID-19 arm than in other arms.

### 3.2.3. COVID-19 outcomes over time

There was a clear trend towards increased prospective registration and blinding over time in COVID-19 trials (figure 3), while there was no clear trend for randomization and use of control arm. The proportion of trials with randomization and use of control arm were highly correlated.

## 3.3. Main analyses

### 3.3.1. Analyses 4 to 7

Analyses 4 to 7 are summarized in figure 4 and values for odds ratios, confidence intervals and *p*-values are given in table 5. Interpretation is given in table 6.

#### 3.3.1.1. *E*-values

*E*-values were calculated for control arm and randomization for both the main direct (analysis 4) and main total (analysis 6) datasets and are presented in table 7. *E*-value interpretation is somewhat subjective, but these values indicate that strong unmeasured confounding would be required to explain the observed effects.

**Table 4.** Descriptive characteristics of entire dataset and stratified by study arm.

| | COVID ($N = 818$) | indication-matched ($N = 839$) | main ($N = 844$) | all ($N = 2501$) |
|---|---|---|---|---|
| control arm | | | | |
| no | 105 (12.8%) | 98 (11.7%) | 211 (25.0%) | 414 (16.6%) |
| yes | 713 (87.2%) | 740 (88.2%) | 632 (74.9%) | 2085 (83.4%) |
| missing | 0 (0%) | 1 (0.1%) | 1 (0.1%) | 2 (0.1%) |
| randomization | | | | |
| no | 41 (5.0%) | 45 (5.4%) | 41 (4.9%) | 127 (5.1%) |
| not applicable | 105 (12.8%) | 98 (11.7%) | 211 (25.0%) | 414 (16.6%) |
| yes | 669 (81.8%) | 693 (82.6%) | 587 (69.5%) | 1949 (77.9%) |
| missing | 3 (0.4%) | 3 (0.4%) | 5 (0.6%) | 11 (0.4%) |
| blinding | | | | |
| no | 424 (51.8%) | 296 (35.3%) | 433 (51.3%) | 1153 (46.1%) |
| yes | 366 (44.7%) | 515 (61.4%) | 387 (45.9%) | 1268 (50.7%) |
| missing | 28 (3.4%) | 28 (3.3%) | 24 (2.8%) | 80 (3.2%) |
| prospective registration | | | | |
| no | 259 (31.7%) | 281 (33.5%) | 222 (26.3%) | 762 (30.5%) |
| yes | 559 (68.3%) | 558 (66.5%) | 622 (73.7%) | 1739 (69.5%) |
| source registry | | | | |
| ChiCTR | 57 (7.0%) | 42 (5.0%) | 60 (7.1%) | 159 (6.4%) |
| CT.gov | 417 (51.0%) | 455 (54.2%) | 352 (41.7%) | 1224 (48.9%) |
| CTRI | 72 (8.8%) | 27 (3.2%) | 63 (7.5%) | 162 (6.5%) |
| EUCTR | 104 (12.7%) | 145 (17.3%) | 182 (21.6%) | 431 (17.2%) |
| IRCT | 109 (13.3%) | 60 (7.2%) | 58 (6.9%) | 227 (9.1%) |
| JPRN | 10 (1.2%) | 45 (5.4%) | 67 (7.9%) | 122 (4.9%) |
| Other | 49 (6.0%) | 65 (7.7%) | 62 (7.3%) | 176 (7.0%) |
| phase | | | | |
| phase 1 | 85 (10.4%) | 86 (10.3%) | 109 (12.9%) | 280 (11.2%) |
| phase 2 | 293 (35.8%) | 181 (21.6%) | 250 (29.6%) | 724 (28.9%) |
| phase 3 | 280 (34.2%) | 253 (30.2%) | 235 (27.8%) | 768 (30.7%) |
| phase 4 | 61 (7.5%) | 186 (22.2%) | 118 (14.0%) | 365 (14.6%) |
| undefined | 93 (11.4%) | 111 (13.2%) | 113 (13.4%) | 317 (12.7%) |
| missing | 6 (0.7%) | 22 (2.6%) | 19 (2.3%) | 47 (1.9%) |
| Africa | | | | |
| no | 770 (94.1%) | 771 (91.9%) | 791 (93.7%) | 2332 (93.2%) |
| yes | 48 (5.9%) | 68 (8.1%) | 53 (6.3%) | 169 (6.8%) |
| North America | | | | |
| no | 649 (79.3%) | 582 (69.4%) | 569 (67.4%) | 1800 (72.0%) |
| yes | 169 (20.7%) | 257 (30.6%) | 275 (32.6%) | 701 (28.0%) |
| Latin America | | | | |
| no | 727 (88.9%) | 760 (90.6%) | 750 (88.9%) | 2237 (89.4%) |
| yes | 91 (11.1%) | 79 (9.4%) | 94 (11.1%) | 264 (10.6%) |
| Asia | | | | |

(Continued.)

| | COVID (*N* = 818) | indication-matched (*N* = 839) | main (*N* = 844) | all (*N* = 2501) |
|---|---|---|---|---|
| no | 458 (56.0%) | 469 (55.9%) | 360 (42.7%) | 1287 (51.5%) |
| yes | 360 (44.0%) | 370 (44.1%) | 484 (57.3%) | 1214 (48.5%) |
| Europe | | | | |
| no | 596 (72.9%) | 542 (64.6%) | 540 (64.0%) | 1678 (67.1%) |
| yes | 222 (27.1%) | 297 (35.4%) | 304 (36.0%) | 823 (32.9%) |
| Oceania | | | | |
| no | 802 (98.0%) | 778 (92.7%) | 759 (89.9%) | 2339 (93.5%) |
| yes | 16 (2.0%) | 61 (7.3%) | 85 (10.1%) | 162 (6.5%) |
| multi-centre | | | | |
| no | 401 (49.0%) | 411 (49.0%) | 401 (47.5%) | 1213 (48.5%) |
| yes | 373 (45.6%) | 377 (44.9%) | 384 (45.5%) | 1134 (45.3%) |
| missing | 44 (5.4%) | 51 (6.1%) | 59 (7.0%) | 154 (6.2%) |
| primary purpose | | | | |
| other | 26 (3.2%) | 101 (12.0%) | 111 (13.2%) | 238 (9.5%) |
| prevention | 121 (14.8%) | 268 (31.9%) | 88 (10.4%) | 477 (19.1%) |
| treatment | 671 (82.0%) | 470 (56.0%) | 645 (76.4%) | 1786 (71.4%) |
| sponsor type | | | | |
| industry | 182 (22.2%) | 261 (31.1%) | 326 (38.6%) | 769 (30.7%) |
| investigator | 52 (6.4%) | 35 (4.2%) | 42 (5.0%) | 129 (5.2%) |
| non industry | 573 (70.0%) | 536 (63.9%) | 433 (51.3%) | 1542 (61.7%) |
| missing | 11 (1.3%) | 7 (0.8%) | 43 (5.1%) | 61 (2.4%) |
| sample size | | | | |
| mean (s.d.) | 893 (5970) | 723 (4820) | 397 (4890) | 668 (5250) |
| median [Q1, Q3] | 100 [50.0, 277] | 120 [45.0, 308] | 80.0 [40.0, 199] | 100 [42.0, 260] |
| missing | 2 (0.2%) | 2 (0.2%) | 0 (0%) | 4 (0.2%) |
| vaccine | | | | |
| no | 771 (94.3%) | 595 (70.9%) | 816 (96.7%) | 2182 (87.2%) |
| yes | 47 (5.7%) | 244 (29.1%) | 28 (3.3%) | 319 (12.8%) |
| conventional | | | | |
| no | 134 (16.4%) | 273 (32.5%) | 76 (9.0%) | 483 (19.3%) |
| yes | 684 (83.6%) | 566 (67.5%) | 768 (91.0%) | 2018 (80.7%) |
| traditional | | | | |
| no | 713 (87.2%) | 776 (92.5%) | 764 (90.5%) | 2253 (90.1%) |
| yes | 105 (12.8%) | 63 (7.5%) | 80 (9.5%) | 248 (9.9%) |
| subject blind | | | | |
| no | 424 (51.8%) | 318 (37.9%) | 416 (49.3%) | 1158 (46.3%) |
| yes | 215 (26.3%) | 268 (31.9%) | 175 (20.7%) | 658 (26.3%) |
| missing | 179 (21.9%) | 253 (30.2%) | 253 (30.0%) | 685 (27.4%) |
| caregiver blind | | | | |
| no | 508 (62.1%) | 420 (50.1%) | 500 (59.2%) | 1428 (57.1%) |
| yes | 131 (16.0%) | 166 (19.8%) | 87 (10.3%) | 384 (15.4%) |

(*Continued.*)

| | COVID ($N = 818$) | indication-matched ($N = 839$) | main ($N = 844$) | all ($N = 2501$) |
|---|---|---|---|---|
| missing | 179 (21.9%) | 253 (30.2%) | 257 (30.5%) | 689 (27.5%) |
| investigator blind | | | | |
| no | 460 (56.2%) | 330 (39.3%) | 458 (54.3%) | 1248 (49.9%) |
| yes | 179 (21.9%) | 256 (30.5%) | 133 (15.8%) | 568 (22.7%) |
| missing | 179 (21.9%) | 253 (30.2%) | 253 (30.0%) | 685 (27.4%) |
| outcome blind | | | | |
| no | 517 (63.2%) | 401 (47.8%) | 485 (57.5%) | 1403 (56.1%) |
| yes | 122 (14.9%) | 189 (22.5%) | 103 (12.2%) | 414 (16.6%) |
| missing | 179 (21.9%) | 249 (29.7%) | 256 (30.3%) | 684 (27.3%) |
| analyst blind | | | | |
| no | 637 (77.9%) | 583 (69.5%) | 587 (69.5%) | 1807 (72.3%) |
| yes | 2 (0.2%) | 3 (0.4%) | 0 (0%) | 5 (0.2%) |
| missing | 179 (21.9%) | 253 (30.2%) | 257 (30.5%) | 689 (27.5%) |

### 3.3.1.2. Geographical regions as confounders

The addition of geographical regions as confounders to the total effect analyses had little impact on results (figure 4 and table 5, analyses labelled 'main total sensitivity (9)' and 'IM total sensitivity (9)').

### 3.3.1.3. Analyse observations without assumed outcomes

Analyses 4–7 were repeated for each outcome with any trials with missing or not applicable values for that outcome removed from analysis (figure 5). Control arm, randomization and blinding had missing or not applicable trials, while prospective registration did not (table 4). Analyses were not therefore repeated for the prospective registration outcome. Results for control arm and blinding were similar to analyses 4 to 7 (figure 4). However, the estimated direct (1.30 [0.64–2.62]) and total (1.21 [0.68–2.15]) effect for randomization in the main dataset was much smaller when trials with not applicable or missing values were excluded from analysis (in the full analyses the values were 1.60 [1.11–2.32] and 1.93 [1.43–2.61], respectively). This is likely because the strong effect on randomization in analyses 4 and 6 was driven by the effect of control arm, as uncontrolled trials were treated as non-randomized.

## 3.4. Exploratory analysis

We did several exploratory analyses using modified DAGs to determine the required adjustment sets. The DAG in the stage 1 IPA (figure 1) assumed that there were no causal relationships between any of the outcome variables. However, we think this assumption is likely false, as noted in a recent commentary [47]. For example, given that we treated uncontrolled studies as not randomized and not blinded, whether the study is controlled has a causal effect on randomization and blinding. This section explores the implications of using an improved DAG.

As per the algorithm outlined by Ankan *et al.* 2021, we tested the implications (conditional independencies) of our DAG with the available data [48]. We specified a theory-driven DAG based on our expectations about the causal relationships between variables. We then applied this DAG to our dataset and tested all the conditional independence statements (d-separation) implied by our theory-driven DAG against the correlation matrix derived from the data. We identified all conditional independence statements that were refuted by the data (95% confidence intervals below −0.05 and above 0.05) and manually added dependencies based on plausible reasoning between relevant variables. Using this iterative process, we updated the theory-driven DAG to resolve conditional independencies that were not supported by the data.

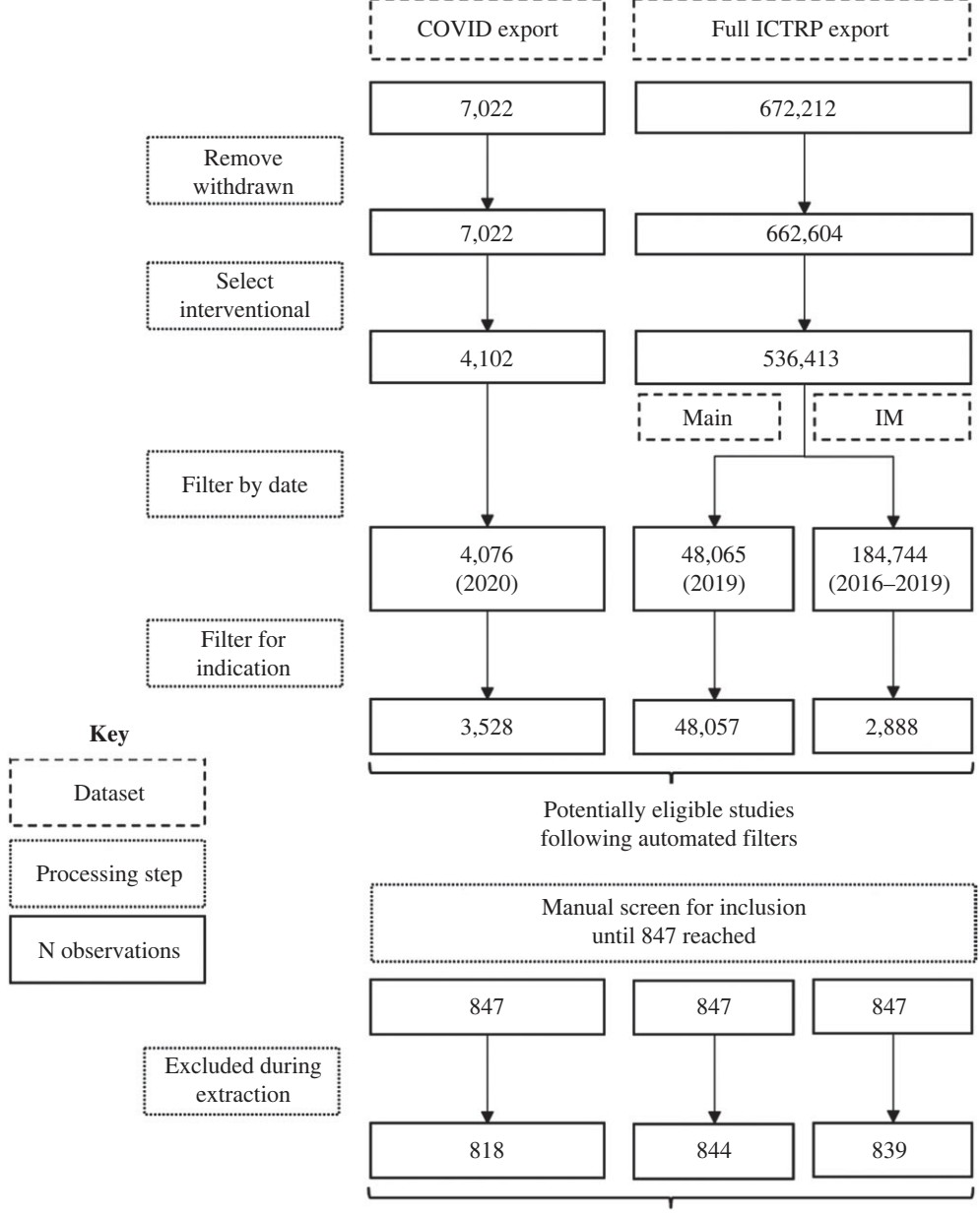

**Figure 2.** Flow of data through the study. COVID export refers to the dataset of COVID-19 trials maintained on the WHO website (https://www.who.int/clinical-trials-registry-platform). Full ICTRP export refers to the full data download available from ICTRP (https://www.who.int/clinical-trials-registry-platform). IM = indication-matched.

Following this process, the minimal adjustment sets for the total effect of all variables remained unchanged (i.e. only sponsor type); however, the minimal adjustment sets for the direct effects differed from our pre-registered plan in important ways. According to the updated DAG, estimating the effect of investigating COVID-19 on randomization and blinding required adjusting for control arm, and other outcome variables. The minimum and sufficient adjustment sets below were identified for each outcome:

— Control arm: sample size, sponsor type, regions, phase, intervention type and primary purpose
— Randomization: sample size, sponsor type, region, control arm, multi-centre, vaccine (i.e. a particular intervention type) and primary purpose
— Blinding: sample size, sponsor type, randomization, region, control arm, phase, intervention type and primary purpose
— Prospective registration: sample size, sponsor type, region, phase and conventional (i.e. a particular intervention type)

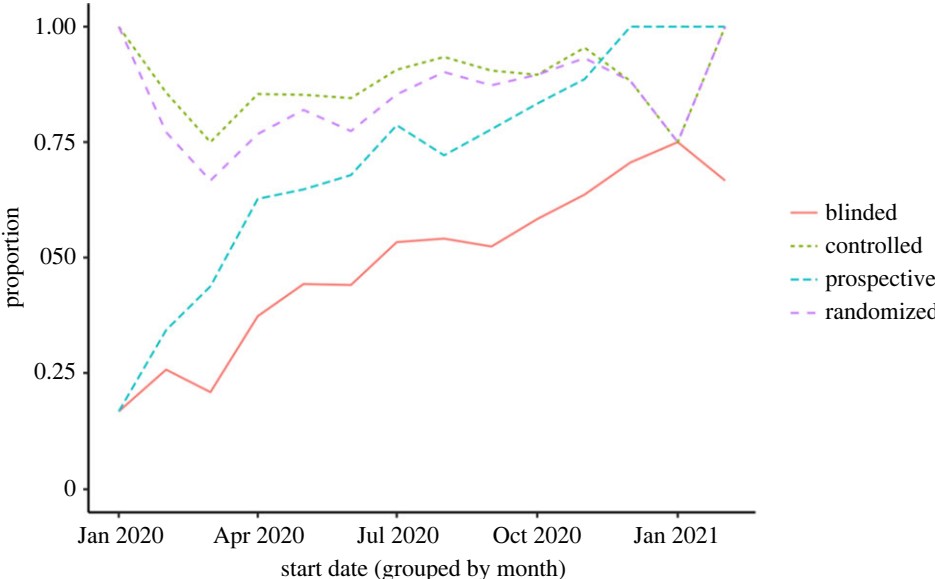

**Figure 3.** COVID-19 trial characteristics over time ($N = 818$ in total, 23 with missing start date excluded from the figure).

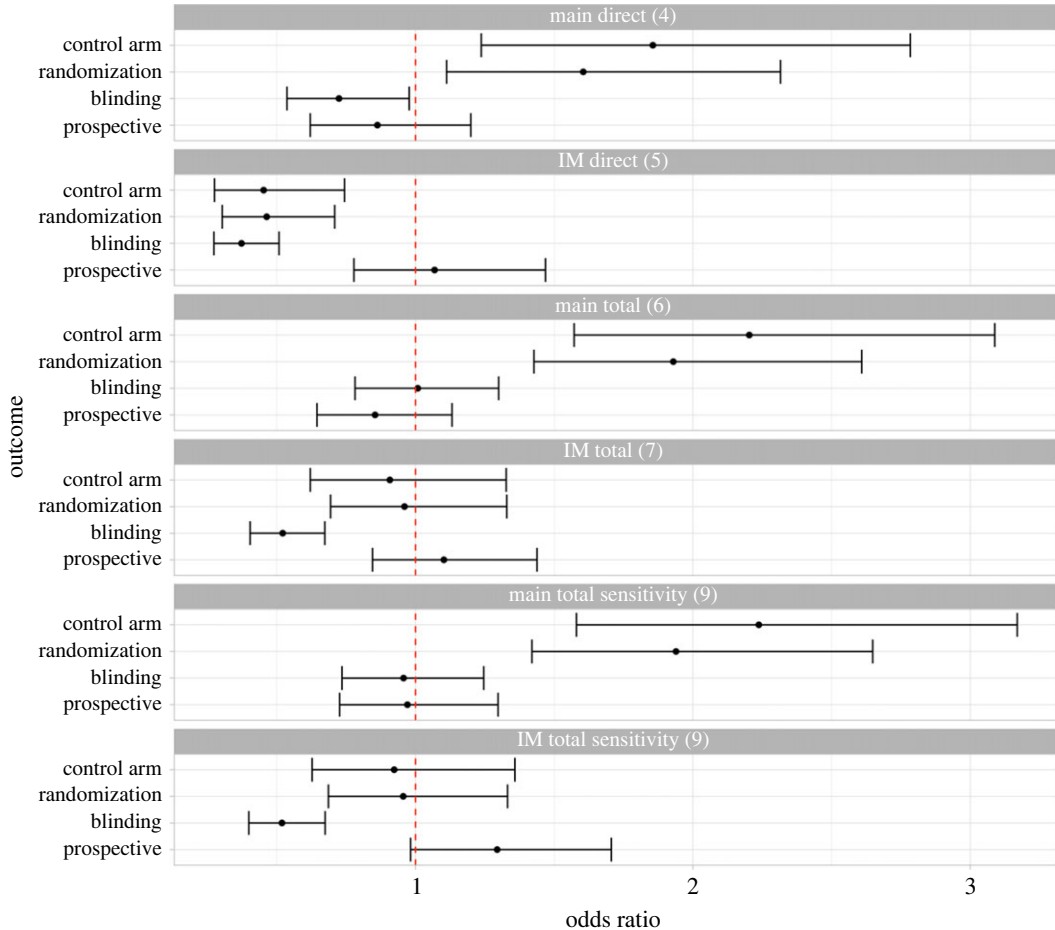

**Figure 4.** Key results figure. Results presented are odds ratios estimated using logistic regression with 95% confidence intervals (Bonferroni corrected). Details of adjustment sets are given in the methods. Numbers in brackets correspond to the analysis numbers given in the methods for ease of reference. 'Main direct' (analysis 4) is the primary analysis. *p*-values and specific values for the odds ratio and confidence interval are given in table 5. IM = indication-matched.

We repeated the primary analysis (analysis 4) using the new adjustment sets. For this exploratory analysis, we used complete cases rather than multiply imputed data for simplicity. For blinding and randomization, inclusion of the control arm variable led to issues with convergence. We therefore

**Table 5.** Full results (IM = indication-matched).

| analysis | outcome | estimate [95% CI] | p-value |
|---|---|---|---|
| main direct (4) | control arm | 1.86 [1.24–2.78] | 0.000146 |
| | randomization | 1.60 [1.11–2.32] | 0.001300 |
| | blinding | 0.72 [0.54–0.98] | 0.007200 |
| | prospective | 0.86 [0.62–1.20] | 0.263000 |
| IM direct (5) | control arm | 0.45 [0.28–0.74] | 0.000071 |
| | randomization | 0.46 [0.30–0.71] | <0.000025 |
| | blinding | 0.37 [0.27–0.51] | <0.000025 |
| | prospective | 1.07 [0.78–1.47] | 0.601000 |
| main total (6) | control arm | 2.20 [1.57–3.09] | <0.000025 |
| | randomization | 1.93 [1.43–2.61] | <0.000025 |
| | blinding | 1.01 [0.782–1.3] | 0.937000 |
| | prospective | 0.85 [0.66–1.13] | 0.162000 |
| IM total (7) | control arm | 0.91 [0.62–1.33] | 0.524000 |
| | randomization | 0.96 [0.69–1.33] | 0.755000 |
| | blinding | 0.52 [0.40–0.67] | <0.000025 |
| | prospective | 1.10 [0.85–1.44] | 0.361000 |
| main total sensitivity (9) | control arm | 2.24 [1.58–3.17] | <0.000025 |
| | randomization | 1.94 [1.42–2.65] | <0.000025 |
| | blinding | 0.96 [0.74–1.25] | 0.677000 |
| | prospective | 0.97 [0.73–1.30] | 0.799000 |
| IM total sensitivity (9) | control arm | 0.92 [0.63–1.36] | 0.605000 |
| | randomization | 0.96 [0.69–1.33] | 0.733000 |
| | blinding | 0.52 [0.40–0.68] | <0.000025 |
| | prospective | 1.29 [0.98–1.71] | 0.019700 |

removed control arm from the adjustment sets. The effect of COVID-19 on control arm (OR: 2.18 [1.60–2.97]), randomization (OR: 1.81 [1.38–2.39]), blinding (OR: 0.57 [0.44–0.75]) and prospective registration (OR: 0.81 [0.64 −1.04]) were qualitatively similar to the primary analysis.

## 4. Discussion

Overall, our findings are inconclusive (figure 4; table 1). The primary analysis generated very strong evidence (per interpretation of p-values in table 1) of a direct and total effect of COVID-19 on the use of control arms and randomization, with increased use of control arms and randomization in COVID-19 trials. The primary analysis also showed weak evidence of a direct effect of COVID-19 on blinding, with decreased use of blinding in COVID-19 trials, and no evidence of a total effect. Use of the indication-matched dataset for analysis, in which trials investigating indications or interventions similar to COVID-19 were included, reversed the direction of the direct effect on use of control arm and randomization. For blinding, in the indication-matched dataset, there was a negative direct and total effect.

In line with our pre-registered interpretation, we conclude that there may be a positive direct effect of COVID-19 on use of control arm and randomization and that there is a positive total effect of COVID-19 on control arm and randomization. We conclude that there is a negative direct effect of COVID-19 on blinding. No evidence of an effect on prospective registration was found.

Sensitivity analyses had mixed results and challenged the findings in some cases. Sensitivity analysis of total effects to include geographical region as a confounder did not change the results in a way that affects interpretation (figure 4; table 1). However, the effect of COVID-19 on randomization was much smaller, and confidence intervals overlapped with one, when trials with missing or not applicable

**Table 6.** Summary results and interpretation.

| question | hypothesis | result | interpretation |
| --- | --- | --- | --- |
| primary aim: direct effect (given figure 1 and assumptions discussed in text) | | | |
| 1. What is the association between investigating COVID-19 and use of control arms? | odds ratio ≠ 1 | There was an effect in the main dataset, but the effect was in the opposite direction in the indication-matched dataset. | There may be a positive association, but it is not robust to differences in control sample. |
| 2. What is the association between investigating COVID-19 and prevalence of randomization? | odds ratio ≠ 1 | There was an effect in the main dataset, but the effect was in the opposite direction in the indication-matched dataset. | There may be a positive association, but it is not robust to differences in control sample. |
| 3. What is the association between investigating COVID-19 and prevalence of at least one form of blinding? | odds ratio ≠ 1 | There was an effect in both the main and indication-matched datasets. | There was an inverse association. |
| 4. What is the association between investigating COVID-19 and prevalence of prospective registration? | odds ratio ≠ 1 | The CI overlapped with 1 in the main dataset and the indication-matched dataset. | There was no evidence of an association. |
| secondary aim: total effect (given figure 1 and assumptions discussed in text) | | | |
| 5. What is the association between investigating COVID-19 and use of control arms? | odds ratio ≠ 1 | There was an effect in the main dataset and the CI overlapped with 1 in the indication-matched dataset. | There was a positive association. |
| 6. What is the association between investigating COVID-19 and prevalence of randomization? | odds ratio ≠ 1 | There was an effect in the main dataset and the CI overlapped with 1 in the indication-matched dataset. | There was a positive association. |
| 7. What is the association between investigating COVID-19 and prevalence of at least one form of blinding? | odds ratio ≠ 1 | The CI overlapped with 1 in the main dataset but there was an effect in the indication-matched dataset. | There was no evidence of an association. |
| 8. What is the association between investigating COVID-19 and prevalence of prospective registration? | odds ratio ≠ 1 | The CI overlapped with 1 in the main dataset and the indication-matched dataset. | There was no evidence of an association. |

values were excluded (figure 5). Additional exploratory analysis with improved adjustment sets were limited by convergence issues but were consistent with the primary analysis.

## 4.1. Comparison to other literature

Our findings contrast with other reports of COVID-19 research quality. Numerous papers have characterized the COVID-19 clinical trial landscape based on trial registrations [32,49–56], generally concluding that COVID-19 studies are of poor quality and that there is substantial room for

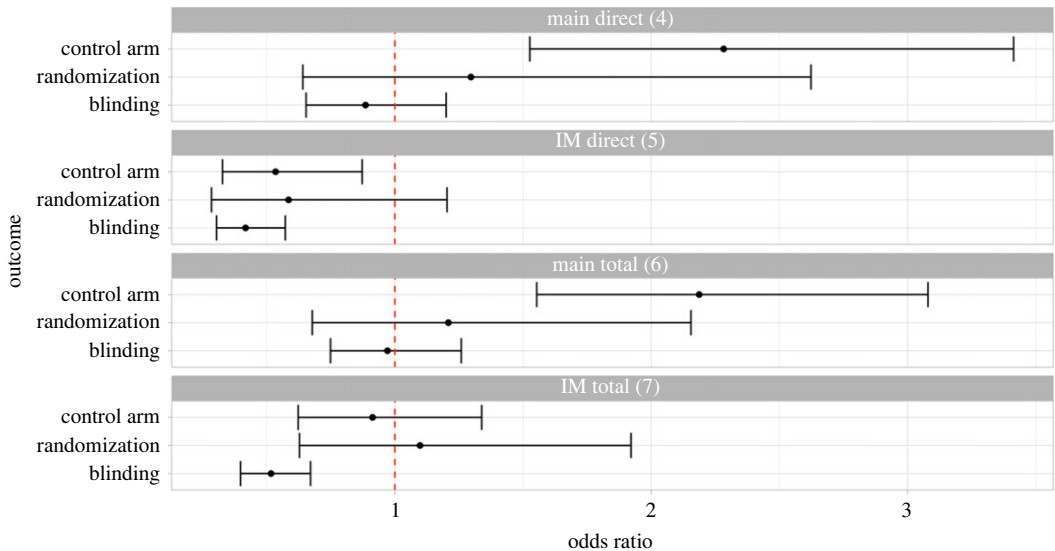

**Figure 5.** Analyses 4 to 7 repeated without trials for which outcome data were missing or not applicable (analysis 10). A table with coefficient, confidence interval and *p*-values is provided in the GitHub repository.

**Table 7.** *E*-values for analyses of the main dataset for which confidence intervals for the odds ratio did not overlap with 1. *E*-values for the point estimate and the lower confidence interval are given.

| analysis | point | lower |
| --- | --- | --- |
| control arm direct effect | 2.32 | 1.73 |
| randomization direct effect | 2.10 | 1.58 |
| control arm total effect | 2.33 | 1.82 |
| randomization total effect | 2.12 | 1.67 |

improvement. However, there was extensive evidence of poor-quality medical research before the pandemic (e.g. summarized in [57]), and relatively few studies have compared COVID-19 trials to pre-pandemic trials. Our study suggests that pandemic research design may not be worse than usual and, based on some outcomes, may be improved.

Some comparative studies have been done, though they suffer from methodological issues. Jung *et al.* [16,58] investigated methodological quality of COVID-19 clinical research in research articles published in journals in comparison to articles from before the pandemic, finding that methodological scores were lower in COVID-19 across all study designs. However, there was no adjustment for confounders or mediators in analysis, making it unclear if COVID-19 was responsible for observed differences. For example, COVID-19 trials, at the time the analysis was done, may have been earlier phase than pre-pandemic trials. The differences observed could therefore be due to phase or other factors that were not specific to COVID-19. Another study investigated the quality of evidence of COVID-19 publications compared to non-COVID-19 papers in high-ranked medical journals [59], similarly finding that COVID-19 articles were of lower quality and also not adjusting for potential confounding or mediating factors. A stage one-registered report comparing the prevalence of statistical reporting inconsistencies in COVID-19 preprints and matched controls has been accepted but results are not yet available [60].

Prevalence of blinding and pre-registration showed a clear increase over time (figure 3). This could indicate a broader trend of improved research quality since the start of the pandemic. If such a trend exists, it may partially explain the divergence between reports of poor-quality COVID-19 research, many of which were published much earlier than this article. However, the lack of trend in randomization and control arm challenges this interpretation.

## 4.2. Strengths

We were explicit about the causal model we assumed when defining our analyses. Although the need to do this in meta-research has been noted [47], as far as we are aware, this is one of the first papers to use a DAG in

a meta-research article. We improved our DAG in exploratory analysis to account for oversights in our pre-registered DAG, in particular by assuming causal effects between outcomes. Many observational meta-research papers do not adjust for other trial design outcomes, which may be an important oversight.

We conducted extensive secondary and sensitivity analyses. The results of these analyses were often not supportive of the primary analysis. Although this complicates interpretation, it highlights that the calculated effect is highly contingent on analytic or design choices for which there is not clear justification. Because this analysis was pre-registered, we fully report all analyses, even those that do not corroborate our primary analysis.

We did duplicate data extraction and duplicate coding of automated data extraction, which should reduce the probability of errors in our dataset. Duplicate manual extraction is common, but automated tasks have the potential to introduce systematic errors and may therefore be even more important than dual manual extraction. Future studies using automated extraction could employ this method.

## 4.3. Limitations

Several limitations are acknowledged in the methods section of the article. Here we focus on limitations that became apparent after stage 1 acceptance. We made several deviations from our pre-registered methods denoted in the text where applicable. These were agreed with the editor but, as some were dependent on observed data, inevitably increase the chance of bias. We could not directly use the pre-registered code because it was not suitable for use on multiple imputed data. We think the risk of bias introduced by this change is very low, as the code used for the multiply imputed datasets is similar to the pre-registered analysis, and the variables adjusted for are identical except where explicitly stated for other reasons. An improved pre-registration would have included analysis code for use on imputed rather than complete data.

In addition, while we examined the clinical trial landscape holistically, we did not examine the impact of individual trials on clinical practice. Certain low-quality COVID-19 trials have had an undue influence on global clinical care [61] and the influence of such trials would not be captured in this study.

### 4.3.1. Analysis plan and study design

We investigated the design of clinical trials in trial registrations which may not accurately reflect trial conduct. Changes may have been made to trial design that are not reflected in the trial registration. It is plausible that the rush to conduct research in the pandemic resulted in more deviations from registered trial design in COVID-19 studies that were not reflected in the registration than is typical outside of a pandemic. If true, analysis of trial registrations may result in COVID-19 studies appearing to be better designed than an analysis of conducted studies would show.

Within each analysis with the full dataset, the effect on randomization and control arm was similar (figure 4). However, when limited to a complete case analysis, the effect on randomization was much smaller (figure 5). This is likely because we treated all uncontrolled trials as non-randomized, and this was not reflected in our DAG specification (figure 1) which assumed that there was no causal effect of any outcome on another. We attempted to address this in exploratory analysis, but convergence issues meant that several analyses with improved adjustment sets could not be run without modification.

### 4.3.2. Data quality

The study is limited by data quality in several ways. Entries for the same trial have been found to be inconsistent across different registries [62]. We relied on the bridging flag variable in the ICTRP export to identify trials cross-registered on other registries for the purpose of determining whether the trial was prospectively registered anywhere; however, it is likely that the bridging flag is incomplete because other trial registrations are not generally well reported in registrations. In many cases, we may not have identified the earliest registered trials, and the proportion of trials prospectively registered may be an underestimate. We do not think that this affects different study arms differently.

We did not attempt to de-duplicate studies that were included more than once in the study due to multiple registrations. Addressing the limitations related to cross-registration would have required substantial manual work which was not feasible. Emerging computational methods are being explored to aid in easing the manual burden of trial matching in future analyses [63].

There are some data quality issues specific to certain registries. For example, the EUCTR 'start date' of trials is problematic [64]. For European Economic Area clinical trial applications, it is the date the study

was authorized to proceed but for trials outside the EAA, it is the date the study was submitted to EudraCT. The registration date in EUCTR is specific to each country with sites used in a trial, rather than reflecting an overall registration date, as used in other registries.

## 5. Conclusion

Our findings are inconclusive, but do not support the findings of related literature. The primary analysis generated very strong evidence of a positive effect of COVID-19 on use of control arms and randomization and weak evidence of a negative effect on blinding. However, secondary and sensitivity analyses challenged the validity of our findings, in some cases reversing the direction of effect. There were also important limitations, including in study design and data quality.

Data accessibility. All code and data (processed and original) are publicly available on the OSF (raw data: https://osf.io/pjc9s/) and GitHub repository (https://github.com/worcjamessmith/COVID-trial-characteristics) and have been archived in Zenodo (https://zenodo.org/record/7396015) [65]. The stage 1 registered report, including the pre-registered code, is available on the OSF: https://osf.io/f6d2v/.

Supplementary material is available online [66].

Authors' contributions. J.A.S.: conceptualization, data curation, formal analysis, investigation, methodology, project administration, supervision, visualization, writing—original draft and writing—review and editing; N.D.: data curation, formal analysis, investigation, methodology and writing—review and editing; H.L.: formal analysis, investigation and writing—review and editing; C.T.: data curation, investigation and writing—review and editing; R.E.A.: data curation, investigation and writing—review and editing; C.K.: conceptualization, data curation, formal analysis, investigation and writing—review and editing.

All authors gave final approval for publication and agreed to be held accountable for the work performed therein.

Conflict of interest declaration. J.A.S. became a consultant to Telis Bioscience/Alvea LLC in January 2022.

Funding. The research was supported by the National Institute for Health Research (NIHR) Oxford Biomedical Research Centre (BRC). The views expressed in this publication are those of the author(s) and not necessarily those of the NHS, the National Institute for Health Research or the department of Health. C.K. is funded by Medical Research Council Population Health Research Unit at the University of Oxford. N.J.D. has received or been employed on funding from the Naji Foundation, The Fetzer Franklin Memorial Fund, The Good Thinking Society, The Laura and John Arnold Foundation, The Mohn Westlake Foundation and The German Federal Ministry of Education and Research.

Acknowledgements. J.A.S. is grateful for the support of National Institute for Health Research (NIHR) Oxford Biomedical Research Council (BRC) and for the support of Prof. Andrew Carr.

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
