## [Peer Review File · Royal Society Open Science]

Review History

RSOS-201543.R0 (Original submission)

Review form: Reviewer 1

Do you have any ethical concerns with this paper?

No

Recommendation?

Accept in principle

Comments to the Author(s)

The scientific validity of the research questions is timely and important.

The logic, rationale, and plausibility of the proposed hypotheses are clear.

On the soundness and feasibility of the methodology and analysis pipeline (including statistical power analysis where applicable): I find it interesting that the authors are choosing to use two different languages (R and Python). R is used for graphing generally (take the numbers and manipulate it) whereas Python is more number based or text based (without the manipulation). Comparing the two might cause discrepancies in the data but I am not sure.

It brings about questions that some researchers and laboratorians are saying currently with the data being shown in studies for testing (that hasn't been peer reviewed). All we have is 1) peer

reviewed or 2) not and decisions will be made on the data. All we can hope for is the labs running the testing to be under some type accrediting body so that quality control and quality assurance is being used. Most mistakes/issues would be found with a good quality system. I find that the clarity and degree of methodological detail is sufficient to replicate the proposed experimental procedures and analysis.

I believe that the authors provide a sufficiently clear and detailed description of the methods to prevent undisclosed flexibility in the experimental procedures or analysis pipeline.

I believe that the authors have considered sufficient outcome-neutral conditions (e.g. positive controls) for ensuring that the results obtained are able to test the stated hypotheses, however, this current environment is rather uncharted territory.

Review form: Reviewer 2 (Florian Naudet)

Do you have any ethical concerns with this paper?

No

Recommendation?

Major revision

Comments to the Author(s)

This is the manuscript for a registered reports that aims to compare the design of clinical research during a pandemic and to compare it with research in non pandemic time.

The protocol reads well and is written using a smart "causal inference" Framework (I must acknowledge that (because I'm rather involved in the field of clinical trials) I'm not totally used with this Framework... so please excuse any stupid question) that aims to explore the effect investigating COVID-19 on prevalence of 4 outcomes related to "risk of bias" (i.e. use of control arms, prevalence of randomisation, blinding and prospective registration). These data will be automatically collected from clinicaltrials.gov.

First, I agree that we need fast assesment of important COVID research that may change clinical practice. I therefore support the initiative of fast tracked registered report that may ensure both sound and fast dissemination of projects that can have an impact on the pandemic. In this case, it is a retrospective an methodological analysis of registrations in clinicaltrials.gov and I'm not sure that it deserves to be fast-tracked. Of course, this is an interesting project but it will not have a direct impact in manadgement of the pandemic. In addition, characteristics of trials may change after this project horizon.

I need to congratulate authors for their methods. I think that authors are doing their best to answer this question. And when asked in causal language, this question is very difficult to adress. Perhaps the best solution would be to compare the 2019 pneumonia trials with the 2020 Covid trials, but it is not possible because there would be too few 2019 pneumonia trials. For this reason authors proposed to perform two analyses that may be complementary :

- A 2019 sample using all type of trials ;
- A 10 year sample using all pneumonias ;

Unfortunately, none of these two sample is perfect and the comparisons may be very prone to confounding. They try to control for main confunders and acknowledge that residual confounding may still be likely.

Among confunders/moderators, I would like to know more about how "time" will be handled. Time might be very related to geographic region, as in the first days of the pandemic there had been a lot of trials in China, and then from Europe, and then from America. One can also see that during time, the type of interventions used have changed. At the beginning we had some trials

about antivirals, and traditional medicine in China for instance, then there has been a massive increase of trials about anti-malaria drugs (e.g. hydroxychloroquine) etc. And I would suspect that the "risk of bias" of any study would be related to the drug/intervention under investigation. Then I also think that the design features explored may change over time. I would suspect that in the first days of the pandemic, one design open label trials easy to implement and then when one has more anticipation, one plan appropriate double blind RCTs. We have seen this pattern during the first 100 days of the pandemic using the covid evidence database. This issue should be anticipated in the present study.

I would like to see more discussion about these conceptual points in the protocol.

Then I would like to see more discussion about the interest of the chosen outcomes.

First, these are somewhat consensual and easy to capture in the database.

But, these outcomes are not perfect, especially without any fine grained analysis:

- A/ Of course, absence of blinding in a RCT is of concern, but the risk of bias highly depends on the outcome explored in the study. I think the risk of bias is not the same for an open label trial exploring chest radiography (that can be subjective) in comparison with an open label trial exploring mortality (e.g. recovery). How can this problem be addressed ?

- B/ I would suggest to label these outcomes 'design characteristics' than "risk of bias" but this is just a suggestion, sometimes there are randomised controlled trials with high risk of bias, and sometimes observational studies with low risk of bias. To me it makes sense to apply Tools to assess risk of bias, for instance in a meta-analysis of RCTs to explore which RCT is more at risk in the context of this meta-analysis. In addition, these Tools are somewhat qualitative and are not binary (e.g. yes/no, as proposed here). In the context of a meta-analysis, it makes sense because all these trials are carefully selected and somewhat similar in terms of question explored. In the context of the larger analyses proposed here and without any qualitative assessment of the trial, I find it difficult to label the 4 design features "risk of bias". I would prefer to label these outcomes as 'key design features'.

- C/ Last, I would be very interested in describing sample size of these trials.

Beside these conceptual points, I have the following remarks :

- As the sample will be heterogeneous, I would suggest to restrict the analysis to 'drugs' and not all interventions. Of course, this is a suggestion open to discussion.

In the introduction, I would suggest to map all existing overlapping initiatives about evidence synthesis for COVID in a more systematic way to demonstrate that none has launched a similar project.

In general, exclusion criteria are not the "negative Mirror" of inclusion criteria : i.e. one cannot exclude a study that was not included. If you include only interventional study, then it is not possible to exclude observational studies. This comment applies to all selection criteria.

In my opinion, there is no sufficient details about missing data :

- Especially for covariates ;

- How these will be handled ?

- Can we have an idea of the completeness of each variable (e.g. using previous works from the team that has an extensive knowledge of clinicaltrials.gov) ;

Can authors write explicitly their regression models ?

Concerning agreement between manual extraction and automatic extraction :

- Is there a threshold that would indicate that automatic extraction are not reliable ;

- I understand that the analysis of pneumonia is a sensitivity analysis but I would prefer to explore agreement on this database on the same number (i.e. 100) of studies than for the 2019 sample. Authors even expect that agreement will be poorer in this database.

To summarise, the protocol is very well written. I made a series of general comments that may improve the discussion of the different assumptions. However, despite this comment I think that there is no better way to explore the question than the way authors proposed. I support the conduct of this study providing the points that I have raised are clarified.

Last, I would make it clear in the protocol how authors will handle any non expected event in the conduct of the study.

Review form: Reviewer 3 (Ioana Cristea)

Do you have any ethical concerns with this paper?

No

Recommendation?

Major revision

Comments to the Author(s)

The authors propose a stage I Registered Report for a retrospective cohort study focused on the quality of clinical trial registrations on COVID-19. (It does look a bit more like a case-control study to me, but as I am (un)fortunately not an epidemiologist, I will defer to more expert reviewers).

Overall, I found the methods and reporting clearly and comprehensively explained, and the authors have offered extensive details about their procedures. The outcome-neutral conditions are judiciously chosen and well-explained. The statistical analysis is painstakingly detailed, including code. A power analysis was explained and attempted, and the authors attempted to compensate for missing key information (ie, what is the expected effect).

However, what I am less convinced about is the relevance and rationale of this research. The authors hypotheses are, simplified, that COVID-19 trials are methodologically worse than trials in non-COVID times. This links to an interesting viewpoint in Science about pandemic exceptionalism.

These are all valid questions, but I think the study proposed it too limited to provide meaningful answers.

First, as shown over and over again across several published papers, clinical trial registrations are often incomplete, containing cursory or even contradictory information. So, using the quality of reporting on a trial registry as a proxy for risk of bias is very limited. A suggestion could be to attempt to retrieve published protocols (often listed with trial registration) and perform risk of bias ratings on those. Protocols generally describe the methods more in detail so at least some domains like randomization, random allocation, blinding, could be rated. This of course takes a lot more manual work, so the authors need to weigh how feasible it is.

Chiefly, I am not convinced of the rationale of pitting all COVID trials versus all non-COVID trials and I don't really see what meaningful conclusion we could draw from this comparison. It does not strike me as particularly informative, ie, what do conclusion can we draw from this result?

Admittedly, COVID trials will include a diversity of interventions, but investigations of trials planned (eg, covid-evidence, covid trial tracked, covid-living NMA etc) so far show they are grouped in a few larger categories, as indications (respiratory, cardiovascular, infectious disease)

and, more granularly, interventions. Non-COVID trials will likely be much more diverse. For example, I doubt there will be many non-inferiority COVID trials, but likely there will be several non-COVID. There will be several non-COVID trials of behavioral interventions (not even mentioned as a separate category in the supplementary table), but very few such COVID trials. I would guess many trials registered in 2019 are probably in oncology, a very research-intensive field. Heterogeneity within the non-COVID cohort is incomparably higher. But more generally, what does it mean that COVID trials are generally worse methodologically (I simplify again)/have higher risk of bias than a mix of non-COVID trials in general?

In my opinion, the authors should aim for an appropriate control cohort, at least attempting to match on some potentially relevant covariates, such as outcome type (eg trials with survival or length hospitalization outcomes), of the conditions for which trials are done (the authors could at least restrict to domains mostly associated with COVID like respiratory, cardiovascular, infectious disease etc), type of intervention and so on. I am not sure how best to do this, but the authors could start from the literature characterizing clinicaltrials.gov in terms of prevalence of trial types, domains, interventions, outcomes and so on and identify the characteristics that would best overlap with COVID. In this sense, I find the sensitivity analysis focused in COVID pneumonia versus non-COVID pneumonia much more informative.

Coding blinding dichotomously is inherently problematic, as blinding of outcome assessors and participants are very distinct things. The former is particularly relevant in relation to type of outcome (ie, generally less consequential for hard outcomes, like death), which while the latter hinges on the intervention (eg, in most behavioral interventions, blinding is not possible). I am not sure how considering them interchangeably for a dichotomous rating is informative.

I am also not convinced that the assumptions of no unmeasured confounding stand and that the only relevant confounder to consider is sponsor (for which the authors make a compelling case). For instance, in non-pandemic settings, it is plausible that trials without a comparator are often trials that the sponsor is using for marketing purposes rather than for actually finding if a treatment works. Or, also likely, for interventions where the sponsor is well-aware that effects of an intervention against an adequate comparator would probably be null. For COVID trials, different rationales might apply, for example lack of a comparator might be the consequence, at least early on, of the difficulty of identifying one such comparator. Treatment as usual, already established treatments and so on are valid comparators for non-COVID trials and hence likely to be used. This is not the case for COVID trials. I don't have expertise in causal modelling to allow me to speculate on how examples such as these affect the authors' inference, if they do. But they definitely speak to the possibility that COVID and non-COVID trials might differ in many ways, including the very assumptions under which they are conducted, and these differences are closely intertwined with risk of bias.

We definitely need to know if the COVID trial literature is mostly a big research waste, but I am not convinced that this research proposal is touching on the most relevant questions in this sense. Two important problems of the COVID literature, identified by the surveys of trials planned so far, are overlap (variations of the same research questions in more trials) and the lack of meaningful outcomes (eg, survival, progression to ICU, duration of ICU stay). Though I agree these issues would probably not be labelled risk of bias, they nonetheless would definitely go under methodological quality. For example, a COVID trial without outcome assessor blinding but a hard outcome (eg survival) is arguably more relevant and of better methodological quality than a trial with outcome assessor blinding but a hard to interpret surrogate outcome.

Review form: Reviewer 4 (Emily Sena)

Do you have any ethical concerns with this paper?

No

Recommendation?

Major revision

Comments to the Author(s)

Smith and colleagues propose to investigate differences between reported risks of bias mitigation in clinical trial registrations of COVID-19 intervention trials versus “other” trials on clinicaltrials.gov.

In principle, I think the proposed study is interesting, considered and well-designed but I do have concerns of some of the decisions and assumptions described below. I hope the author’s find this of value.

1. I must admit that I am not an expert in trial registries but this study will assess the reporting quality at the time of registration which presents a challenge of the fidelity of the registration to the actual trial design (that you discuss). I suspect this may have been looked at in the past but you can envisage that due to the pressures of the pandemic that registrations are not robustly or comprehensively completed and a likely moderator of the reporting quality will be the completeness of the entry itself.
2. The choice to restrict to only clinicaltrials.gov appears to be driven by the desire to automate the data collection rather than that clinicaltrials.gov is a representative sample of clinical trial registries. A quick look on [covid19trialstracker](https://covid19trialstracker.com) (although the data haven’t been updated since 12 August) suggests that almost half the COVID-19 trials are not registered on clinicaltrials.gov. This is an important consideration that could substantially limit the inferences made by the study and a shame given the existence of ICTRP albeit extra manual work.
3. Have the authors considered that existing trials that were registered prior to COVID may have adapted their design to add COVID participants? And if this has any implications?
4. I understand the decision to use a historical sample for the control group as fewer trials are likely to have been registered and I suspect practice is unlikely to have deviated in 1 year but this is an assumption that should probably be checked? The number of Non-COVID intervention registered in 2020 should at least be presented as this would be the obvious comparator group.
5. I was interested in your choice to look at “at least one form of blinding”? My understanding is that trial entries state whether the design is single or double-blinded. Why is this difference not considered?
6. For the pneumonia sample, I understand that you have chosen a longer period to ensure the sample is sufficiently large but do you have any rules to prioritise the inclusion more recent data once you reach your desired sample size?
7. I would encourage the authors to consider a more robust data management plan for manual extraction. It is very easy in Excel to remove, delete or accidentally add data and because of the lack of an audit trail, you may never know this has happened, especially when dealing with large datasets. Systematic review tools are freely available, amenable to this type of research and may provide a useful and robust alternative.
8. Sensitivity of trial identification – the ICTRP search portal has also created csv/xml outputs for COVID trials. This may be an alternative data source one-step less processed than the

Decision letter (RSOS-201543.R0)

Dear Dr Smith,

The Editors assigned to your Stage 1 Registered Report ("Association between COVID-19 investigation and risk of bias in clinical trial registrations on clinicaltrials.gov: a retrospective cohort study") have now received comments from reviewers. We would like you to revise your paper in accordance with the referee and editors suggestions which can be found below (not including confidential reports to the Editor). Please note this decision does not guarantee eventual acceptance.

Please submit a copy of your revised paper within three months (i.e. by the 26-Sep-2020). If we do not hear from you within this time then it will be assumed that the paper has been withdrawn. In exceptional circumstances, extensions may be possible if agreed with the Editorial Office in advance. We do not allow multiple rounds of revision so we urge you to make every effort to fully address all of the comments at this stage. If deemed necessary by the Editors, your manuscript will be sent back to one or more of the original reviewers for assessment. If the original reviewers are not available we may invite new reviewers.

When submitting your revised manuscript, you must respond to the comments made by the referees and upload a file "Response to Referees" in "Section 2 - File Upload". Please use this to document how you have responded to the comments, and the adjustments you have made. In order to expedite the processing of the revised manuscript, please be as specific as possible in your response.

Kind regards,
Professor Chris Chambers
Royal Society Open Science
openscience@royalsociety.org

on behalf of Professor Chris Chambers (Registered Reports Editor, Royal Society Open Science)
openscience@royalsociety.org

Associate Editor Comments to Author (Professor Chris Chambers):
Comments to the Author:

Four specialist reviewers have now assessed the Stage 1 manuscript, and at the outset I would like to extend a big thank you to the reviewers for the very rapid and high quality reviews. As you will see, all of the reviewers find merit in the proposal, but Reviewers 2, 3 and 4 also raise some substantial issues that will need to be addressed to achieve Stage 1 acceptance. Headlines include the effects of confounders, validity of the coding methodology and assumptions, limitations driven by a focus on clinicaltrials.gov and basic registration information (cf. additional registries and full protocols), appropriateness of the comparison group (in turn limiting the degree of insight yielding by the study), and the degree of methodological detail.

I have also read the manuscript, and in light of these assessments I am convinced overall that the study has substantial value in principle, and is within scope for Stage 1 revision. However, I would caution the authors that achieving Stage 1 in-principle acceptance will require significant work to satisfy the concerns raised. Reviewer 2 notes that the submission need not be fast-tracked; at RSOS, all COVID-related RRs submitted as part of this initiative are fast-tracked by

default, but would stress that this need not imply the need for haste on the authors' end. Should the authors decide to revise, please take sufficient time respond comprehensively to all points raised.

Comments to Author:

Reviewer: 1

Comments to the Author(s)

The scientific validity of the research questions is timely and important.

The logic, rationale, and plausibility of the proposed hypotheses are clear.

On the soundness and feasibility of the methodology and analysis pipeline (including statistical power analysis where applicable): I find it interesting that the authors are choosing to use two different languages (R and Python). R is used for graphing generally (take the numbers and manipulate it) whereas Python is more number based or text based (without the manipulation).

Comparing the two might cause discrepancies in the data but I am not sure.

It brings about questions that some researchers and laboratorians are saying currently with the data being shown in studies for testing (that hasn't been peer reviewed). All we have is 1) peer reviewed or 2) not and decisions will be made on the data. All we can hope for is the labs running the testing to be under some type accrediting body so that quality control and quality assurance is being used. Most mistakes/issues would be found with a good quality system.

I find that the clarity and degree of methodological detail is sufficient to replicate the proposed experimental procedures and analysis.

I believe that the authors provide a sufficiently clear and detailed description of the methods to prevent undisclosed flexibility in the experimental procedures or analysis pipeline.

I believe that the authors have considered sufficient outcome-neutral conditions (e.g. positive controls) for ensuring that the results obtained are able to test the stated hypotheses, however, this current environment is rather uncharted territory.

Reviewer: 2

Comments to the Author(s)

This is the manuscript for a registered reports that aims to compare the design of clinical research during a pandemic and to compare it with research in non pandemic time.

The protocol reads well and is written using a smart "causal inference" Framework (I must acknowledge that (because I'm rather involved in the field of clinical trials) I'm not totally used with this Framework... so please excuse any stupid question) that aims to explore the effect investigating COVID-19 on prevalence of 4 outcomes related to "risk of bias" (i.e. use of control arms, prevalence of randomisation, blinding and prospective registration). These data will be automatically collected from clinicaltrials.gov.

First, I agree that we need fast assesment of important COVID research that may change clinical practice. I therefore support the initiative of fast tracked registered report that may ensure both sound and fast dissemination of projects that can have an impact on the pandemic. In this case, it is a retrospective an methodological analysis of registrations in clinicaltrials.gov and I'm not sure that it deserves to be fast-tracked. Of course, this is an interesting project but it will not have a direct impact in manadgement of the pandemic. In addition, characteristics of trials may change after this project horizon.

I need to congratulate authors for their methods. I think that authors are doing their best to answer this question. And when asked in causal language, this question is very difficult to adress. Perhaps the best solution would be to compare the 2019 pneumonia trials with the 2020 Covid trials, but it is not possible because there would be too few 2019 pneumonia trials. For this reason authors proposed to perform two analyses that may be complementary :

- A 2019 sample using all type of trials ;
- A 10 year sample using all pneumonias ;

Unfortunately, none of these two sample is perfect and the comparisons may be very prone to confounding. They try to control for main confounders and acknowledge that residual confounding may still be likely.

Among confounders/moderators, I would like to know more about how "time" will be handled. Time might be very related to geographic region, as in the first days of the pandemic there had been a lot of trials in China, and then from Europe, and then from America. One can also see that during time, the type of interventions used have changed. At the beginning we had some trials about antivirals, and traditional medicine in China for instance, then there has been a massive increase of trials about anti-malaria drugs (e.g. hydroxychloroquine) etc. And I would suspect that the "risk of bias" of any study would be related to the drug/intervention under investigation. Then I also think that the designs features explore may change across time. I would suspect that in the first day of the pandemic, one design open label trials easy to implement and then when one has more anticipation, one plan appropriate double blind RCTs. We have seen this pattern during the 100 first day of the pandemic using the covid evidence database. This issue should be anticipated in the present study.

I would like to see more discussion about these conceptual points in the protocol.

Then I would like to see more discussion about the interest of the chosen outcomes.

First, these are somewhat consensual and easy to capture in the database.

But, these outcome are not perfect, especially without any fine grained analysis:

- A/ Of course, absence of blinding in a RCT is of concern, but the risk of bias highly depends on the outcome explored in the study. I think the risk of bias is not the same for an open label trial exploring chest radiography (that can be subjective) in comparison with an open label trial exploring mortality (e.g. recovery). How can this problem be addressed ?

- B/ I would suggest to label these outcomes 'designs characteristics' than "risk of bias" but this is just a suggestion, sometimes there are randomised controlled trials with high risk of bias, and sometimes observational studies with low risk of bias. To me it makes sense to apply Tools to assess risk of bias, for instance in a meta-analysis of RCTs to explore which RCT is more at risk in the context of this meta-analysis. In addition, these Tools are somewhat qualitative and are not binary (e.g. yes/no, as proposed here). In the context of a meta-analysis, it makes sense because all these trials are carefully selected and somewhat similar in terms of question explored. In the context of the larger analyses proposed here and without any qualitative assessment of the trial, I find it difficult to label the 4 designs features "risk of bias". I would prefer to label these outcomes are 'key design features'.

- C/ Last, I would be very interested in describing sample size of these trials.

Beside these conceptual points, I have the following remarks :

- As the sample will be heterogeneous, I would suggest to restrict the analysis to 'drugs' and not all interventions. Of course, this is a suggestion open to discussion.

In the introduction, I would suggest to map all existing overlapping initiatives about evidence synthesis for COVID in a more systematic way to demonstrate that none has launched a similar project.

In general, exclusion criteria are not the "negative Mirror" of inclusion criteria : i.e. one cannot exclude a study that was not include. If you include only interventional study, then it is not possible to exclude observational studies. This comment applies to all selection criteria.

In my opinion, there is no sufficient details about missing data :

- Especially for covariates ;

- How these will be handled ?

- Can we have an idea of the completeness of each variable (e.g. using previous works from the team that has an extensive knowledge of clinicaltrials.gov) ;

Can authors write explicitly their regression models ?

Concerning agreement between manual extraction and automatic extraction :

- Is there a threshold that would indicate that automatic extraction are not reliable ;
- I understand that the analysis of pneumonia is a sensitivity analysis but I would prefer to explore agreement on this database on the same number (i.e. 100) of studies than for the 2019 sample. Authors even expect that agreement will be poorer in this database.

To summarise, the protocol is very well written. I made a series of general comments that may improve the discussion of the different assumptions. However, despite this comment I think that there is no better way to explore the question than the way authors proposed. I support the conduct of this study providing the points that I have raised are clarified.

Last, I would make it clear in the protocol how authors will handle any non expected event in the conduct of the study.

Reviewer: 3

Comments to the Author(s)

The authors propose a stage I Registered Report for a retrospective cohort study focused on the quality of clinical trial registrations on COVID-19. (It does look a bit more like a case-control study to me, but as I am (un)fortunately not an epidemiologist, I will defer to more expert reviewers).

Overall, I found the methods and reporting clearly and comprehensively explained, and the authors have offered extensive details about their procedures. The outcome-neutral conditions are judiciously chosen and well-explained. The statistical analysis is painstakingly detailed, including code. A power analysis was explained and attempted, and the authors attempted to compensate for missing key information (ie, what is the expected effect).

However, what I am less convinced about is the relevance and rationale of this research. The authors hypotheses are, simplified, that COVID-19 trials are methodologically worse than trials in non-COVID times. This links to a interesting viewpoint in Science about pandemic exceptionalism.

These are all valid questions, but I think the study proposed it too limited to provide meaningful answers.

First, as shown over and over again across several published papers, clinical trial registrations are often incomplete, containing cursory or even contradictory information. So, using the quality of reporting on a trial registry as a proxy for risk of bias is very limited. A suggestion could be to attempt to retrieve published protocols (often listed with trial registration) and perform risk of bias ratings on those. Protocols generally describe the methods more in detail so at least some domains like randomization, random allocation, blinding, could be rated. This of course takes a lot more manual work, so the authors need to weigh how feasible it is.

Chiefly, I am not convinced of the rationale of pitting all COVID trials versus all non-COVID trials and I don't really see what meaningful conclusion we could draw from this comparison. It does not strike me as particularly informative, ie, what do conclusion can we draw from this result?

Admittedly, COVID trials will include a diversity of interventions, but investigations of trials planned (eg, covid-evidence, covid trial tracked, covid-living NMA etc) so far show they are grouped in a few larger categories, as indications (respiratory, cardiovascular, infectious disease) and, more granularly, interventions. Non-COVID trials will likely be much more diverse. For example, I doubt there will be many non-inferiority COVID trials, but likely there will be several non-COVID. There will be several non-COVID trials of behavioral interventions (not even

mentioned as a separate category in the supplementary table), but very few such COVID trials. I would guess many trials registered in 2019 are probably in oncology, a very research-intensive field. Heterogeneity within the non-COVID cohort is incomparably higher. But more generally, what does it mean that COVID trials are generally worse methodologically (I simplify again)/have higher risk of bias than a mix of non-COVID trials in general?

In my opinion, the authors should aim for an appropriate control cohort, at least attempting to match on some potentially relevant covariates, such as outcome type (eg trials with survival or length hospitalization outcomes), of the conditions for which trials are done (the authors could at least restrict to domains mostly associated with COVID like respiratory, cardiovascular, infectious disease etc), type of intervention and so on. I am not sure how best to do this, but the authors could start from the literature characterizing clinicaltrials.gov in terms of prevalence of trial types, domains, interventions, outcomes and so on and identify the characteristics that would best overlap with COVID. In this sense, I find the sensitivity analysis focused in COVID pneumonia versus non-COVID pneumonia much more informative.

Coding blinding dichotomously is inherently problematic, as blinding of outcome assessors and participants are very distinct things. The former is particularly relevant in relation to type of outcome (ie, generally less consequential for hard outcomes, like death), which while the latter hinges on the intervention (eg, in most behavioral interventions, blinding is not possible). I am not sure how considering them interchangeably for a dichotomous rating is informative.

I am also not convinced that the assumptions of no unmeasured confounding stand and that the only relevant confounder to consider is sponsor (for which the authors make a compelling case). For instance, in non-pandemic settings, it is plausible that trials without a comparator are often trials that the sponsor is using for marketing purposes rather than for actually finding if a treatment works. Or, also likely, for interventions where the sponsor is well-aware that effects of an intervention against an adequate comparator would probably be null. For COVID trials, different rationales might apply, for example lack of a comparator might be the consequence, at least early on, of the difficulty of identifying one such comparator. Treatment as usual, already established treatments and so on are valid comparators for non-COVID trials and hence likely to be used. This is not the case for COVID trials. I don't have expertise in causal modelling to allow me to speculate on how examples such as these affect the authors' inference, if they do. But they definitely speak to the possibility that COVID and non-COVID trials might differ in many ways, including the very assumptions under which they are conducted, and these differences are closely intertwined with risk of bias.

We definitely need to know if the COVID trial literature is mostly a big research waste, but I am not convinced that this research proposal is touching on the most relevant questions in this sense. Two important problems of the COVID literature, identified by the surveys of trials planned so far, are overlap (variations of the same research questions in more trials) and the lack of meaningful outcomes (eg, survival, progression to ICU, duration of ICU stay). Though I agree these issues would probably not be labelled risk of bias, they nonetheless would definitely go under methodological quality. For example, a COVID trial without outcome assessor blinding but a hard outcome (eg survival) is arguably more relevant and of better methodological quality than a trial with outcome assessor blinding but a hard to interpret surrogate outcome.

Reviewer: 4

Comments to the Author(s)

Smith and colleagues propose to investigate differences between reported risks of bias mitigation in clinical trial registrations of COVID-19 intervention trials versus "other" trials on clinicaltrials.gov.

In principle, I think the proposed study is interesting, considered and well-designed but I do have concerns of some of the decisions and assumptions described below. I hope the author's find this of value.

1. I must admit that I am not an expert in trial registries but this study will assess the reporting quality at the time of registration which presents a challenge of the fidelity of the registration to the actual trial design (that you discuss). I suspect this may have been looked at in the past but you can envisage that due to the pressures of the pandemic that registrations are not robustly or comprehensively completed and a likely moderator of the reporting quality will be the completeness of the entry itself.
2. The choice to restrict to only clinicaltrials.gov appears to be driven by the desire to automate the data collection rather than that clinicaltrials.gov is a representative sample of clinical trial registries. A quick look on covid19trialstracker (although the data haven't been updated since 12 August) suggests that almost half the COVID-19 trials are not registered on clinicaltrials.gov. This is an important consideration that could substantially limit the inferences made by the study and a shame given the existence of ICTRP albeit extra manual work.
3. Have the authors considered that existing trials that were registered prior to COVID may have adapted their design to add COVID participants? And if this has any implications?
4. I understand the decision to use a historical sample for the control group as fewer trials are likely to have been registered and I suspect practice is unlikely to have deviated in 1 year but this is an assumption that should probably be checked? The number of Non-COVID intervention registered in 2020 should at least be presented as this would be the obvious comparator group.
5. I was interested in your choice to look at "at least one form of blinding"? My understanding is that trial entries state whether the design is single or double-blinded. Why is this difference not considered?
6. For the pneumonia sample, I understand that you have chosen a longer period to ensure the sample is sufficiently large but do you have any rules to prioritise the inclusion more recent data once you reach your desired sample size?
7. I would encourage the authors to consider a more robust data management plan for manual extraction. It is very easy in Excel to remove, delete or accidentally add data and because of the lack of an audit trail, you may never know this has happened, especially when dealing with large datasets. Systematic review tools are freely available, amenable to this type of research and may provide a useful and robust alternative.
8. Sensitivity of trial identification – the ICTRP search portal has also created csv/xml outputs for COVID trials. This may be an alternative data source one-step less processed than the

Author's Response to Decision Letter for (RSOS-201543.R0)

See Appendix A.

RSOS-201543.R1 (Revision)

Review form: Reviewer 2 (Florian Naudet)

Do you have any ethical concerns with this paper?

No

Recommendation?

Accept in principle

Comments to the Author(s)

Authors made a good job in addressing my questions. I think the manuscript is ready now for an in principle acceptance.

Review form: Reviewer 3 (Ioana Cristea)

Do you have any ethical concerns with this paper?

No

Recommendation?

Accept in principle

Comments to the Author(s)

The authors have addressed all my comments, substantially expanded their study and justified all their choices. I look forward to the results and have no further comments.

Decision letter (RSOS-201543.R1)

Dear Dr Smith

On behalf of the Editor, I am pleased to inform you that your Manuscript RSOS-201543.R1 entitled "Association between investigating COVID-19 and design characteristics in global clinical trial registrations" has been accepted in principle for publication in Royal Society Open Science. The reviewers' and editors' comments are included at the end of this email.

Please read the following email carefully

Your accepted Stage 1 manuscript has been publicly registered at:
<https://doi.org/10.17605/OSF.IO/5YAEB>

You may now progress to Stage 2 and complete the study as approved.

Following completion of your study, we invite you to resubmit your paper for peer review as a Stage 2 Registered Report. Please note that your manuscript can still be rejected for publication at Stage 2 if the Editors consider any of the following conditions to be met:

- The results were unable to test the authors' proposed hypotheses by failing to meet the approved outcome-neutral criteria.
- The authors altered the Introduction, rationale, or hypotheses, as approved in the Stage 1 submission.
- The authors failed to adhere closely to the registered experimental procedures. Please note that any deviations from the approved experimental procedures must be communicated to the editor immediately for approval, and prior to the completion of data collection. Failure to do so can result in revocation of in-principle acceptance and rejection at Stage 2 (see complete guidelines for further information).
- Any post-hoc (unregistered) analyses were either unjustified, insufficiently caveated, or overly dominant in shaping the authors' conclusions.
- The authors' conclusions were not justified given the data obtained.

We encourage you to read the complete guidelines for authors concerning Stage 2 submissions at <https://royalsocietypublishing.org/rsos/registered-reports#ReviewerGuideRegRep>. Please especially note the requirements for data sharing, reporting the URL of the independently registered protocol, and that withdrawing your manuscript will result in publication of a Withdrawn Registration.

Once again, thank you for submitting your manuscript to Royal Society Open Science and we look forward to receiving your Stage 2 submission. If you have any questions at all, please do not hesitate to get in touch. We look forward to hearing from you shortly with the anticipated submission date for your stage two manuscript.

on behalf of Professor Chris Chambers (Registered Reports Editor, Royal Society Open Science)
openscience@royalsociety.org

Associate Editor Comments to Author (Professor Chris Chambers):

Two of the four original reviewers were available to assess the revised manuscript. Both agree that the revised manuscript thoroughly addresses the concerns raised in round 1, and on this basis they recommend Stage 1 IPA.

Reviewers' comments to Author:

Reviewer: 2

Comments to the Author(s)

Authors made a good job in addressing my questions. I think the manuscript is ready now for an in principle acceptance.

Reviewer: 3

Comments to the Author(s)

The authors have addressed all my comments, substantially expanded their study and justified all their choices. I look forward to the results and have no further comments.

RSOS-201543.R2 (Revision)

Review form: Reviewer 1 (Cathleen Hanlon)

Is the manuscript scientifically sound in its present form?

Yes

Are the interpretations and conclusions justified by the results?

Yes

Is the language acceptable?

Yes

Do you have any ethical concerns with this paper?

No

Have you any concerns about statistical analyses in this paper?

No

Recommendation?

Accept as is

Comments to the Author(s)

The manuscript is fine as it is currently written.

Review form: Reviewer 2 (Florian Naudet)

Is the manuscript scientifically sound in its present form?

Yes

Are the interpretations and conclusions justified by the results?

Yes

Is the language acceptable?

Yes

Do you have any ethical concerns with this paper?

No

Have you any concerns about statistical analyses in this paper?

No

Recommendation?

Accept with minor revision

Comments to the Author(s)

Florian Naudet, MD, PhD, Rennes 1 University

. Whether the data are able to test the authors' proposed hypotheses by passing the approved outcome-neutral criteria (such as absence of floor and ceiling effects or success of positive controls)

YES

. Whether the Introduction, rationale and stated hypotheses are the same as the approved Stage 1 submission

YES

. Whether the authors adhered precisely to the registered experimental procedures

YES with the following precision: authors did not fully adhere to the original protocol but all the changes were agreed on with the editor and are tracked in the manuscript. It is therefore 100 % fine.

. Where applicable, whether any unregistered exploratory statistical analyses are justified, methodologically sound, and informative

YES

. Whether the authors' conclusions are justified given the data

YES.

I have the following minor (cosmetic) suggestions for change :

- p13. l23 : please provide a link to the histogram that identifies implausible values ;
- for the models (e.g. p.14-15) should we add a term for residuals (+ ϵ) ?
- Figure 2 appears twice ;
- Figure 3 : I would be interested to have both the percentages and the numbers at each time point or at least confidence intervals at each time point for each numbers / the current figure could be a little bit difficult to follow and perhaps a little bit misleading ;
- Figure 4 and Table 5 must be merged in the same figure (e.g. like forest plots in meta-analyses that present numerical values) ;
- Table 6 point 5-6 and 7. I don't understand why the interpretation does not make it clear that the positive associations were not robust regarding the differences in control samples;
- p32 l.7 : "E value interpretation is..." this is in my opinion an interpretation and should be moved to the discussion as it should not appear in the results ;
- First line of the discussion : "our results are inconclusive" is sufficient. I don't think that it is possible to be "inconclusive" and "to point toward a positive effect..." at the same time. I agree that the interpretation is inconclusive and I would therefore delete the discussion about "positive effects".
- Regarding the comparison with other literature, I would like to see here a discussion of reference 16 that is presented earlier in the manuscript.
- In the conclusion, I would not start by stating that these results contradict the previous literature but rather insist on the fact that these results are inconclusive. I really prefer in terms of interpretation the last sentence : "our results are inconclusive but do not support the findings of the related literature". I would also edit the abstract in the same spirit in order to avoid any spin : please delete "the finding contradict. much existing..." and replace with "the findings do not support the existing...".
- Please also add a few words about limitations in the abstract in order to avoid any spin.

Kudos for this great registered report. I support its publication.

Decision letter (RSOS-201543.R2)

Dear Dr Smith:

On behalf of the Editor, I am pleased to inform you that your Stage 2 Registered Report RSOS-201543.R2 entitled "Estimating the effect of COVID-19 on trial design characteristics: a registered report" has been deemed suitable for publication in Royal Society Open Science subject to minor revision in accordance with the referee suggestions. Please find the referees' comments at the end of this email.

The reviewers and Subject Editor have recommended publication, but also suggest some minor revisions to your manuscript. We invite you to respond to the comments and revise your manuscript. Below the referees' and Editors' comments (where applicable) we provide additional requirements. Final acceptance of your manuscript is dependent on these requirements being met. We provide guidance below to help you prepare your revision.

Please submit your revised manuscript and required files (see below) no later than 7 days from today's (ie 01-Nov-2022) date. Note: the ScholarOne system will 'lock' if submission of the

revision is attempted 7 or more days after the deadline. If you do not think you will be able to meet this deadline please contact the editorial office immediately.

on behalf of Professor Chris Chambers
(Registered Reports Editor, Royal Society Open Science)
openscience@royalsociety.org

Associate Editor Comments to Author (Professor Chris Chambers):

Associate Editor: 1

Comments to the Author:

At outset, thank you for your patience awaiting this interim decision. I had between expecting comments from one additional reviewer, but they are now very late and I have decided to proceed without their input.

Two of the reviewers from Stage 1 were available to assess your completed Stage 2 manuscript. As you will see, their comments are extremely positive, and very much mirror my own assessment. This is a fine exemplar of the RR format, tackling an important and timely question.

Reviewer 2 offers some helpful suggestions for minor revisions in the interests of clarity. Once these are addressed, final acceptance will be awarded without further review.

Comments to Author:

Reviewer: 2

Comments to the Author(s)

Florian Naudet, MD, PhD, Rennes 1 University

. Whether the data are able to test the authors' proposed hypotheses by passing the approved outcome-neutral criteria (such as absence of floor and ceiling effects or success of positive controls)

YES

. Whether the Introduction, rationale and stated hypotheses are the same as the approved Stage 1 submission

YES

. Whether the authors adhered precisely to the registered experimental procedures

YES with the following precision: authors did not fully adhere to the original protocol but all the changes were agreed on with the editor and are tracked in the manuscript. It is therefore 100 % fine.

. Where applicable, whether any unregistered exploratory statistical analyses are justified, methodologically sound, and informative

YES

. Whether the authors' conclusions are justified given the data

YES.

I have the following minor (cosmetic) suggestions for change :

- p13. l23 : please provide a link to the histogram that identifies implausible values ;
- for the models (e.g. p.14-15) should we add a term for residuals (+ ϵ) ?
- Figure 2 appears twice ;
- Figure 3 : I would be interested to have both the percentages and the numbers at each time point or at least confidence intervals at each time point for each numbers / the current figure could be a little bit difficult to follow and perhaps a little bit misleading ;
- Figure 4 and Table 5 must be merged in the same figure (e.g. like forest plots in meta-analyses that present numerical values) ;
- Table 6 point 5-6 and 7. I don't understand why the interpretation does not make it clear that the positive associations were not robust regarding the differences in control samples;
- p32 l.7 : "E value interpretation is..." this is in my opinion an interpretation and should be moved to the discussion as it should not appear in the results ;
- First line of the discussion : "our results are inconclusive" is sufficient. I don't think that it is possible to be "inconclusive" and "to point toward a positive effect..." at the same time. I agree that the interpretation is inconclusive and I would therefore delete the discussion about "positive effects".
- Regarding the comparison with other literature, I would like to see here a discussion of reference 16 that is presented earlier in the manuscript.
- In the conclusion, I would not start by stating that these results contradict the previous literature but rather insist on the fact that these results are inconclusive. I really prefer in terms of interpretation the last sentence : "our results are inconclusive but do not support the findings of the related literature". I would also edit the abstract in the same spirit in order to avoid any spin : please delete "the finding contradict. much existing..." and replace with "the findings do not support the existing...".
- Please also add a few words about limitations in the abstract in order to avoid any spin.

Kudos for this great registered report. I support its publication.

Reviewer: 1

Comments to the Author(s)

The manuscript is fine as it is currently written.

===PREPARING YOUR MANUSCRIPT===

one version should clearly identify all the changes that have been made (for instance, in coloured highlight, in bold text, or tracked changes);

===PREPARING YOUR REVISION IN SCHOLARONE===

- If you are providing image files for potential cover images, please upload these at this step, and inform the editorial office you have done so. You must hold the copyright to any image provided.
- A copy of your point-by-point response to referees and Editors. This will expedite the preparation of your proof.

- Ensure that your data access statement meets the requirements at <https://royalsociety.org/journals/authors/author-guidelines/#data>. You should ensure that you cite the dataset in your reference list. If you have deposited data etc in the Dryad repository, please only include the 'For publication' link at this stage. You should remove the 'For review' link.
- If you are requesting an article processing charge waiver, you must select the relevant waiver option (if requesting a discretionary waiver, the form should have been uploaded, see 'File upload' above).
- If you have uploaded any electronic supplementary (ESM) files, please ensure you follow the guidance at <https://royalsociety.org/journals/authors/author-guidelines/#supplementary-material> to include a suitable title and informative caption. An example of appropriate titling and captioning may be found at https://figshare.com/articles/Table_S2_from_Is_there_a_trade-off_between_peak_performance_and_performance_breadth_across_temperatures_for_aerobic_scope_in_teleost_fishes_/3843624.

Author's Response to Decision Letter for (RSOS-201543.R2)

See Appendix B.

Decision letter (RSOS-201543.R3)

Dear Dr Smith:

I am pleased to inform you that your manuscript entitled "Estimating the effect of COVID-19 on trial design characteristics: a registered report" is now accepted for publication in Royal Society Open Science.

Please remember to make any data sets or code libraries 'live' prior to publication, and update any links as needed when you receive a proof to check - for instance, from a private 'for review' URL to a publicly accessible 'for publication' URL. It is also good practice to add data sets, code and other digital materials to your reference list.

COVID-19 rapid publication process:

We are taking steps to expedite the publication of research relevant to the pandemic. If you wish, you can opt to have your paper published as soon as it is ready, rather than waiting for it to be published the scheduled Wednesday.

This means your paper will not be included in the weekly media round-up which the Society sends to journalists ahead of publication. However, it will still appear in the COVID-19 Publishing Collection which journalists will be directed to each week (<https://royalsocietypublishing.org/topic/special-collections/novel-coronavirus-outbreak>).

If you wish to have your paper considered for immediate publication, or to discuss further, please notify openscience_proofs@royalsociety.org and press@royalsociety.org when you respond to this email.

Royal Society Open Science is a fully open access journal. A payment may be due before your article is published. Our partner Copyright Clearance Center's RightsLink for Scientific Communications will contact the corresponding author about your open access options from the email domain @copyright.com (if you have any queries regarding fees, please see <https://royalsocietypublishing.org/rsos/charges> or contact authorfees@royalsociety.org).

on behalf of Professor Chris Chambers (Subject Editor).

Follow Royal Society Publishing on Twitter: @RSocPublishing
Follow Royal Society Publishing on Facebook:
<https://www.facebook.com/RoyalSocietyPublishing/>
Read Royal Society Publishing's blog:
<https://royalsociety.org/blog/blogsearchpage/?category=Publishing>

Appendix A

Line-by-line responses to referees

Associate Editor Comments to Author (Professor Chris Chambers):

Comments to the Author:

Four specialist reviewers have now assessed the Stage 1 manuscript, and at the outset I would like to extend a big thank you to the reviewers for the very rapid and high quality reviews. As you will see, all of the reviewers find merit in the proposal, but Reviewers 2, 3 and 4 also raise some substantial issues that will need to be addressed to achieve Stage 1 acceptance. Headlines include the effects of confounders, validity of the coding methodology and assumptions, limitations driven by a focus on clinicaltrials.gov and basic registration information (cf. additional registries and full protocols), appropriateness of the comparison group (in turn limiting the degree of insight yielding by the study), and the degree of methodological detail.

I have also read the manuscript, and in light of these assessments I am convinced overall that the study has substantial value in principle, and is within scope for Stage 1 revision. However, I would caution the authors that achieving Stage 1 in-principle acceptance will require significant work to satisfy the concerns raised. Reviewer 2 notes that the submission need not be fast-tracked; at RSOS, all COVID-related RRs submitted as part of this initiative are fast-tracked by default, but would stress that this need not imply the need for haste on the authors' end. Should the authors decide to revise, please take sufficient time respond comprehensively to all points raised.

Thank you for these comments and for the opportunity to submit a revised manuscript. The manuscript has been changed considerably since initial submission, and we therefore use this space to provide a summary and explanation of the major changes to the protocol that may be useful for all reviewers to read. There have been three major developments.

First, we conducted a pilot study involving manual extraction of 50 COVID-19 and 50 non-COVID-19 trials from the International Clinical Trials Registry Platform (ICTRP) data export. We aimed to explore a number of suggestions made by reviewers, primarily the feasibility of expanding our dataset to include trials outside of clinicaltrials.gov. We therefore did not include trials registered on clinicaltrials.gov. We found that most data we needed were feasible to manually extract, and that the prospects for automating some data extraction were better than we had anticipated. The full write-up of the pilot study and accompanying datasets are available here:

https://osf.io/pjc9s/?view_only=1314dcb40c3640009e10fca85ba7d7aa.

Second, therefore, rather than relying on clinicaltrials.gov data, we have decided to expand our sample to trials from the ICTRP, which provides access to a database of trials from 18 trial registries internationally and therefore gives a more representative sample of global trials. However, because the use of this data source will result in considerably more manual work per trial, we now limit the sample size of our analysis to be that required by the power calculation, whereas before we planned to collect data on all available COVID-19 trials. This is a trade-off that we think is more than worth it to improve the generalizability of our investigation and avoid the biases may arise from using only clinicaltrials.gov data. As a

consequence of using ICTRP, we are unfortunately no longer able to include the covariate “sponsor experience” in our analysis, because this is a variable calculable only from clinicaltrials.gov, though we are still able to collect ‘sponsor type’ to adjust for in analysis. Several of the quality checks previously proposed are no longer relevant with this dataset and have been removed or replaced. We now include an additional covariate in analysis: source registry.

Third, we have expanded the second dataset (previously the pneumonia dataset) to include more indications and meet the sample size required by our sample size calculation for the main analysis. One reviewer (reviewer 3) felt strongly that we needed to limit the inclusion criteria for the control arm of our main analysis; we disagree and prefer to use a random sample for our main analysis for the reasons explained in the responses, but appreciate that others might value the indication-matched analysis. The increased sample size should allow better checks on the robustness of our findings to differences in the control arm used. This dataset will also be sampled from ICTRP and will use the same COVID-19 trials as the main dataset.

Another minor change which is not made in response to any specific comment is that for the purposes of analysis we now treat single group trials as non-randomised, whereas before we only included multi-group trials analysis of randomisation. This is in line with other literature^{1,2} and ensures that the sample size will be sufficient for that outcome. On reflection, it is also an accurate representation of the outcome, as, clearly, single arm trials are non-randomised.

We are grateful for all the feedback received and hope that we have adequately addressed all the concerns. We think that our new protocol represents a significant improvement on the first version and hope that the reviewers agree.

Comments to Author:

Reviewer: 1

Comments to the Author(s)

The scientific validity of the research questions is timely and important.

Thank you for this comment.

The logic, rationale, and plausibility of the proposed hypotheses are clear.

Thank you for this comment.

On the soundness and feasibility of the methodology and analysis pipeline (including statistical power analysis where applicable): I find it interesting that the authors are choosing to use two different languages (R and Python). R is used for graphing generally (take the numbers and manipulate it) whereas Python is more number based or text based (without the manipulation). Comparing the two might cause discrepancies in the data but I am not sure.

Thank you for this comment. Since we have defined the variables and values that will be extracted from the data, we are confident that the same results should in principle be

achievable using the two different methods, even if the exact route to reach those results differs, for example due to differences in the language. If there are discrepancies, this will indicate that there is a problem with one of the methods and will be addressed.

It brings about questions that some researchers and laboratorians are saying currently with the data being shown in studies for testing (that hasn't been peer reviewed). All we have is 1) peer reviewed or 2) not and decisions will be made on the data. All we can hope for is the labs running the testing to be under some type accrediting body so that quality control and quality assurance is being used. Most mistakes/issues would be found with a good quality system.

Thank you for this comment.

I find that the clarity and degree of methodological detail is sufficient to replicate the proposed experimental procedures and analysis.

Thank you for this comment.

I believe that the authors provide a sufficiently clear and detailed description of the methods to prevent undisclosed flexibility in the experimental procedures or analysis pipeline.

Thank you for this comment.

I believe that the authors have considered sufficient outcome-neutral conditions (e.g. positive controls) for ensuring that the results obtained are able to test the stated hypotheses, however, this current environment is rather uncharted territory.

Thank you for this comment.

Reviewer: 2

Comments to the Author(s)

This is the manuscript for a registered reports that aims to compare the design of clinical research during a pandemic and to compare it with research in non pandemic time.

The protocol reads well and is written using a smart "causal inference" Framework (I must acknowledge that (because I'm rather involved in the field of clinical trials) I'm not totally used with this Framework... so please excuse any stupid question) that aims to explore the effect investigating COVID-19 on prevalence of 4 outcomes related to "risk of bias" (i.e. use of control arms, prevalence of randomisation, blinding and prospective registration). These data will be automatically collected from clinicaltrials.gov.

First, I agree that we need fast assesment of important COVID research that may change clinical practice. I therefore support the initiative of fast tracked registered report that may ensure both sound and fast dissemination of projects that can have an impact on the pandemic. In this case, it is a retrospective an methodological analysis of registrations in clinicaltrials.gov and I'm not sure that it deserves to be fast-tracked. Of course, this is an interesting project but it will not have a direct impact in manadgement of the pandemic. In addition, characteristics of trials may change after this project horizon.

I need to congratulate authors for their methods. I think that authors are doing their best to answer this question. And when asked in causal language, this question is very difficult to address. Perhaps the best solution would be to compare the 2019 pneumonia trials with the 2020 Covid trials, but it is not possible because there would be too few 2019 pneumonia trials. For this reason authors proposed to perform two analyses that may be complementary :

- A 2019 sample using all type of trials ;
- A 10 year sample using all pneumonias ;

Unfortunately, none of these two sample is perfect and the comparisons may be very prone to confounding. They try to control for main confounders and acknowledge that residual confounding may still be likely.

Thank you for these comments.

Among confounders/moderators, I would like to know more about how "time" will be handled. Time might be very related to geographic region, as in the first days of the pandemic there had been a lot of trials in China, and then from Europe, and then from America. One can also see that during time, the type of interventions used have changed. At the beginning we had some trials about antivirals, and traditional medicine in China for instance, then there has been a massive increase of trials about anti-malaria drugs (e.g. hydroxychloroquine) etc. And I would suspect that the "risk of bias" of any study would be related to the drug/intervention under investigation. Then I also think that the design features explore may change across time. I would suspect that in the first day of the pandemic, one design open label trials easy to implement and then when one has more anticipation, one plan appropriate double blind RCTs. We have seen this pattern during the 100 first day of the pandemic using the covid evidence database. This issue should be anticipated in the present study.

I would like to see more discussion about these conceptual points in the protocol.

Thank you for this comment. We agree that trial characteristics may be related to time since the start of the pandemic, and think that this change is worth assessing during analysis of our data (analysis section 3). Based on emerging literature, the reviewer's observations about changes in trial locations and treatments over time seem to be correct³. However, we do not think that time needs to be considered as a confounder or mediator in the models. The question we are asking is, in general, whether the characteristics of COVID-19 trials compared to non-COVID-19 trials differ, over the time frame from which we are able to sample. If there was a difference, but most of the lower quality COVID-19 trials were conducted at the beginning of the pandemic, we would still conclude that there was an association between COVID-19 and trial design characteristics – it just may be that an explanation for this is that there was a rush to plan and design trials early in the pandemic. Our protocol includes descriptive analysis of trial design over time in the COVID-19 arm. If our descriptive analysis indicates changes, these will be discussed in our paper and exploratory analysis could be conducted to shed light on whether the conclusions would change depending on the time of the pandemic that is sampled. Since such analysis would be exploratory we have not included discussion of it in the stage one report.

We will also be adjusting for trial location and have added a sub-category of alternative medicine for adjustment during analysis to address some of the concerns around intervention type.

Then I would like to see more discussion about the interest of the chosen outcomes. First, these are somewhat consensual and easy to capture in the database. But, these outcomes are not perfect, especially without any fine grained analysis:

- A/ Of course, absence of blinding in a RCT is of concern, but the risk of bias highly depends on the outcome explored in the study. I think the risk of bias is not the same for an open label trial exploring chest radiography (that can be subjective) in comparison with an open label trial exploring mortality (e.g. recovery). How can this problem be addressed ?

Thank you for this comment. We agree that the risk of bias from lack of blinding depends on various other aspects, such as the outcome type or who exactly is blinded. At the recommendation of this reviewer (see next comment and response), we have changed the terminology in the manuscript to "design characteristics" rather than "risk of bias" because it more accurately reflects what we are measuring. We have added further discussion about the limitations in using the blinding outcome, at the point where it is introduced. The text now reads:

"There are limitations in treating blinding dichotomously as an outcome. For example, the importance of blinding may depend on the subjectivity of the outcome, and who exactly is blinded. There is also uncertainty about the importance of blinding in trials⁴. We have included this outcome because we feel that most assessments of trial quality would include it, but it should be interpreted in the context of these limitations. Where possible we will collect information on who was blinded in the trial and report these data. For the main analysis, we will treat blinding as dichotomous because we expect details of blinding to be missing in some cases (pilot study and refs: ^{5,6})."

- B/ I would suggest to label these outcomes 'design characteristics' than "risk of bias" but this is just a suggestion, sometimes there are randomised controlled trials with high risk of bias, and sometimes observational studies with low risk of bias. To me it makes sense to apply Tools to assess risk of bias, for instance in a meta-analysis of RCTs to explore which RCT is more at risk in the context of this meta-analysis. In addition, these Tools are somewhat qualitative and are not binary (e.g. yes/no, as proposed here). In the context of a meta-analysis, it makes sense because all these trials are carefully selected and somewhat similar in terms of question explored. In the context of the larger analyses proposed here and without any qualitative assessment of the trial, I find it difficult to label the 4 design features "risk of bias". I would prefer to label these outcomes as 'key design features'.

Thank you for this comment. We agree with the reviewer's rationale that it may be more reasonable to call the outcomes design characteristics. We have therefore changed the terminology throughout the protocol and now introduce these outcomes as trial characteristics often considered in risk of bias assessments.

- C/ Last, I would be very interested in describing sample size of these trials.

Thank you for this comment. We will also extract information on the sample size of the trials, and will include this as a covariate in our model.

Beside these conceptual points, I have the following remarks :

- As the sample will be heterogeneous, I would suggest to restrict the analysis to 'drugs' and not all interventions. Of course, this is a suggestion open to discussion.

Thank you for this suggestion, which is in line with a suggestion made also by reviewer 3. On reflection, we agree that restricting the analysis to a more homogenous sample of only drugs will improve the comparability of the samples and will therefore do this.

In the introduction, I would suggest to map all existing overlapping initiatives about evidence synthesis for COVID in a more systematic way to demonstrate that none has launched a similar project.

Thank you for this comment. There are a large number of evidence synthesis efforts for COVID-19 underway, making a full systematic mapping of them a research project in itself which is unfortunately beyond the scope of this study. However, to examine if anyone has launched a similar project, we conducted a targeted, reproducible search of PubMed literature. Specifically, we searched for studies using clinical trial registry information to compare a sample of COVID-19 trials to non-COVID-19 trials in terms of risk of bias or design characteristics. We have included details of this in the manuscript. The text now reads:

“To identify other published studies investigating similar research questions using trial registries, we searched PubMed (supplementary material for search details) for studies published from 1st March 2020 to 27th October 2020 and Open Science Framework for relevant project registrations (no date restrictions). Two authors (JAS and CT) reviewed the results. We also reviewed the related research section of COVID-evidence (<https://covid-evidence.org/related-research>) and generally searched the internet. No directly comparable studies were identified. We are aware of one preprint investigating a similar question based on journal articles rather than trial registrations⁷.”

In general, exclusion criteria are not the "negative Mirror" of inclusion criteria : i.e. one cannot exclude a study that was not include. If you include only interventional study, then it is not possible to exclude observational studies. This comment applies to all selection criteria.

Thank you for this comment. We have now updated the inclusion/exclusion criteria to avoid any redundancy. The criteria now read:

Inclusion and exclusion criteria

All datasets

Inclusion criteria

- Interventional trials, as determined by the study type specified in the ICTRP data export

- At least one arm of the trial investigating a drug (including small molecules, antibodies, proteins, blood-derived products (e.g. plasma), biologicals, ATMPs, vaccines, herbal therapies and vitamin therapies)
- For the COVID-19 arm:
 - o Investigating a condition including the term “COVID-19” (or any of the synonyms listed in supplementary material) in the listed “condition” in the ICTRP export
 - o First posted from 1st January 2020

Exclusion criteria:

- Trials with status of withdrawn, defined as “Study halted prematurely, prior to enrollment of first participant”²⁴.
- Studies where the intervention is combined (aka drugs given alongside another intervention)
- Homeopathic treatments
- For control arms:
 - o Trials whose recruitment criteria has been adapted to include COVID-19 patients

Main dataset control arm

Inclusion criteria:

- Investigating any condition other than COVID-19 as defined above
- Registration date from 1st January 2019 to the same date in 2019 that we download the COVID-19 sample on in 2020

Indication-matched dataset control arm

Inclusion criteria:

- Investigating a condition including one of the following terms in the ICTRP export but not COVID-19 as defined above:
 - o septic shock
 - o multi organ failure; multiple organ failure, multiple organ dysfunction syndrome, multiple systems organ failure, multisystem organ failure
 - o cardiogenic shock
 - o myocarditis, myocardial inflammation
 - o myocardial ischaemia
 - o respiratory failure, respiratory insufficiency
 - o ARDS, respiratory distress syndrome
 - o Pneumonia
 - o influenza, flu
 - o respiratory arrest, apnea, breathing cessation, breathing stops, pulmonary arrest
 - o cardiac arrest, heart attack, asystole, asystolia, asystolic
- Registration date from 1st January 2018 to 31st December 2019

In my opinion, there is no sufficient details about missing data :

- Especially for covariates ;
- How these will be handled ?

- Can we have an idea of the completeness of each variable (e.g. using previous works from the team that has an extensive knowledge of clinicaltrials.gov) ;

Thank you for this comment. With clinicaltrials.gov data we did not expect there to be missingness in the variables as defined, and therefore did not include details of handling missing data. However, with the expansion of our dataset to include ICTRP data, it is likely that there will be missing data. For missing covariates we will use multiple imputation by chained equations to impute values. Based on our pilot study, we expect most variables to be well reported, though phase in particular was not well reported in the Chinese trial registry. The table from our pilot study is provided below for reference (Table 1) and the full pilot study report is available here:

https://osf.io/pjc9s/?view_only=1314dcb40c3640009e10fca85ba7d7aa. Note that these proportions are applicable to non-clinicaltrials.gov data only. In clinicaltrials.gov we expect there to be almost no missingness. Therefore, the eventual proportions of missing data will be much lower than those presented, as clinicaltrials.gov trials comprise ~50% of all trials in ICTRP.

Table 1: Proportion of unreported data for each covariate in COVID-19 and non-COVID-19 clinical trial registrations

Covariate	Proportion of Unreported Data (%)	
	Non-COVID-19 (n=50)	COVID-19 (n=50)
Target number	4	0
Primary purpose	4	2
Multi-Centre	12	8
Phase of Trial	64	26
Sponsor Type	0	0
Geographic region	0	0

Can authors write explicitly their regression models ?

Thank you for this comment. We have now included the regression models in the paper. They are also specified in code in the supplementary analysis. The paper now reads:

Primary analysis

4. Main dataset: direct effect

Our primary analysis will present the association (i.e. odds ratio) of investigating COVID-19 on each outcome in the main dataset, conditional on all measured covariates: i.e. the direct effect of COVID-19, given the interpretation discussed above.

Specifically, for each of the four outcomes the model will be:

$$\begin{aligned}
 \text{Logit}(\text{outcome} \mid \text{covariates}) = & \beta_0 + \beta_1 \text{COVID} & \text{Eq. 1} \\
 & + \beta_2 \text{Source registry} + \beta_3 \text{Phase} + \beta_4 \text{Region} \\
 & + \beta_5 \text{Multicentre} + \beta_6 \text{Intervention type} \\
 & + \beta_7 \text{Primary purpose} + \beta_8 \text{Sponsor type} \\
 & + \beta_9 \text{Sample size} + \beta_{10} \text{Intervention type}
 \end{aligned}$$

where each variable is of the type and values specified in the supplementary data dictionary.

Secondary analyses

5. Indication-matched dataset: direct effect

Analysis 4 (Main dataset: direct effect) will be repeated for the indication-matched dataset (Eq. 1).

6. Main dataset: total effect

We will present the association (i.e. odds ratio) of investigating COVID-19 on each outcome in the main dataset adjusting only for sponsor type, which was identified as a potential confounder: i.e. the total effect of COVID-19, given the interpretation discussed above.

Specifically, for each of the four outcomes the model will be:

$$\text{Logit}(\text{outcome} \mid \text{covariates}) = \beta_0 + \beta_1 \text{COVID} + \beta_2 \text{Sponsor type} \quad \text{Eq. 2}$$

7. Indication-matched dataset: total effect

Analysis 6 (Main dataset: total effect) will be repeated for the indication-matched dataset (Eq. 2).

Sensitivity analysis

8. E-values

If the confidence intervals of the odds ratios for COVID-19 in the analyses of the main dataset (analysis 4 and 6) do not overlap with one, we will calculate e-values^{9,10} for the odds ratios and confidence interval closest to one. The calculation of e-values for odds ratios relies on an approximate conversion of odds ratios to relative risk (specifically, the relative risk is calculated as the square root of the odds ratio¹¹). We will provide the calculated relative risk and the e-value for the point estimate and confidence interval closest to the null.

9. Geographic regions as confounders

We will repeat the total effect analyses (analyses 6 and 7), adjusting for geographic region as well as sponsor type, given that there is some uncertainty in the relationship between those variables and investigating COVID-19.

Specifically, for each of the four outcomes the model will be:

$$\begin{aligned}
 \text{Logit}(\text{outcome} \mid \text{covariates}) = & \beta_0 + \beta_1 \text{COVID} + \beta_2 \text{Sponsor type} & \text{Eq. 3} \\
 & + \beta_3 \text{Region}
 \end{aligned}$$

Concerning agreement between manual extraction and automatic extraction :

- Is there a threshold that would indicate that automatic extraction are not reliable ;
- I understand that the analysis of pneumonia is a sensitivity analysis but I would prefer to explore agreement on this database on the same number (i.e. 100) of studies than for the 2019 sample. Authors even expect that agreement will be poorer in this database.

Thank you for this comment. We have specified in outcome neutral criterion 1, "To rely on automated extraction for a variable within a registry, we will require that there is ultimately >95% agreement between the automated and manual extraction for that variable and registry." If the agreement is lower than that threshold, we will consider the automatic extraction to not be sufficiently reliable.

The pneumonia analysis has been replaced by an analysis including more indications. We will manually check the same proportion of that dataset (15%) as the main dataset.

To summarise, the protocol is very well written. I made a series of general comments that may improve the discussion of the different assumptions. However, despite this comment I think that there is no better way to explore the question than the way authors proposed. I support the conduct of this study providing the points that I have raised are clarified.

Thank you for this comment.

Last, I would make it clear in the protocol how authors will handle any non expected event in the conduct of the study.

Thank you for this comment. As required in registered reports at RSOS, any unanticipated events that require a change to our protocol will be discussed with the editorial board immediately. We have not included this in our protocol since it is a requirement for all articles of this type.

Reviewer: 3

Comments to the Author(s)

The authors propose a stage I Registered Report for a retrospective cohort study focused on the quality of clinical trial registrations on COVID-19. (It does look a bit more like a case-control study to me, but as I am (un)fortunately not an epidemiologist, I will defer to more expert reviewers).

Overall, I found the methods and reporting clearly and comprehensively explained, and the authors have offered extensive details about their procedures. The outcome-neutral conditions are judiciously chosen and well-explained. The statistical analysis is painstakingly detailed, including code. A power analysis was explained and attempted, and the authors attempted to compensate for missing key information (ie, what is the expected effect).

Thank you for this comment.

However, what I am less convinced about is the relevance and rationale of this research. The authors hypotheses are, simplified, that COVID-19 trials are methodologically worse than trials in non-COVID times. This links to a interesting viewpoint in Science about pandemic exceptionalism.

These are all valid questions, but I think the study proposed it too limited to provide meaningful answers.

Thank you for this comment. We hope we have addressed and/or responded to the specific points adequately below.

First, as shown over and over again across several published papers, clinical trial registrations are often incomplete, containing cursory or even contradictory information. So, using the quality of reporting on a trial registry as a proxy for risk of bias is very limited. A suggestion could be to attempt to retrieve published protocols (often listed with trial registration) and perform risk of bias ratings on those. Protocols generally describe the methods more in detail so at least some domains like randomization, random allocation, blinding, could be rated. This of course takes a lot more manual work, so the authors need to weigh how feasible it is.

Thank you for this comment and the suggestion to explore protocols rather than rely on information in trial registrations. To explore the feasibility of using protocols, we filtered the interventional COVID-19 trials on clinicaltrials.gov by those that have a study protocol posted (https://clinicaltrials.gov/ct2/results?cond=COVID-19&age_v=&qndr=&type=Intr&rslt=&u_prot=Y&Search=Apply). On 7th October, only 36 studies had such a protocol. Recently published literature also notes that they “rarely identified protocols or manuscripts, precluding more detailed analyses of trial designs”³. As such, we do not think using protocols is feasible.

While we agree that trial registrations are not a perfect source of information, we are using them to gather information about trial characteristics, rather than to examine quality of reporting in the registry. In terms of completeness, in our pilot study each of the outcomes was well reported.

Regarding the fact that they may contain contradictory information, in the section “Outcome neutral criteria and quality control”, number 3: “Identify inconsistencies in outcome information in trial registrations”, explicitly aims to quantify this, and states that “If there is >10% contradictory information for any outcome, we will try to determine which information sources are accurate and correct the data”. We also feel that using registry entries, which may not contain as much information as protocols or publications, is a worthwhile trade-off because the population of studies available for analysis is not distorted by the publication process and is therefore more representative of the population of studies that was actually planned.

We also note that there are a large number of published papers using clinical trial registries as a data source that could not feasibly be answered using papers or protocols.

Taken together, we are confident that these concerns are addressed in the protocol and that trial registries are, although imperfect, a suitable source of information.

Chiefly, I am not convinced of the rationale of pitting all COVID trials versus all non-COVID trials and I don't really see what meaningful conclusion we could draw from this comparison. It does not strike me as particularly informative, ie, what do conclusion can we

draw from this result?

Admittedly, COVID trials will include a diversity of interventions, but investigations of trials planned (eg, covid-evidence, covid trial tracked, covid-living NMA etc) so far show they are grouped in a few larger categories, as indications (respiratory, cardiovascular, infectious disease) and, more granularly, interventions. Non-COVID trials will likely be much more diverse. For example, I doubt there will be many non-inferiority COVID trials, but likely there will be several non-COVID. There will be several non-COVID trials of behavioral interventions (not even mentioned as a separate category in the supplementary table), but very few such COVID trials. I would guess many trials registered in 2019 are probably in oncology, a very research-intensive field. Heterogeneity within the non-COVID cohort is incomparably higher. But more generally, what does it mean that COVID trials are generally worse methodologically (I simplify again)/have higher risk of bias than a mix of non-COVID trials in general?

In my opinion, the authors should aim for an appropriate control cohort, at least attempting to match on some potentially relevant covariates, such as outcome type (eg trials with survival or length hospitalization outcomes), of the conditions for which trials are done (the authors could at least restrict to domains mostly associated with COVID like respiratory, cardiovascular, infectious disease etc), type of intervention and so on. I am not sure how best to do this, but the authors could start from the literature characterizing clinicaltrials.gov in terms of prevalence of trial types, domains, interventions, outcomes and so on and identify the characteristics that would best overlap with COVID. In this sense, I find the sensitivity analysis focused in COVID pneumonia versus non-COVID pneumonia much more informative.

We are grateful for these comments, and recognise that limiting heterogeneity in the non-COVID cohort could be a useful way to improve the planned study. A similar point was also made by reviewer 2, who recommended focussing only on drugs to limit heterogeneity. At this stage, it is difficult to fully assess heterogeneity and how it may differ between cohorts. However, some historical data are available^{2,12} and it is plausible that there will be differences in terms of interventions and there will certainly be differences in indications. We have decided to limit the study to focus only on drugs as a result of these comments.

We feel, however, that meaningful conclusions can be drawn from the comparison across all indications. If we find a difference between COVID-19 and other studies, we would be able to conclude that, at least in the design characteristics we look at, the design of COVID-19 trials differs from the design of trials planned outside of the pandemic. If we accept that the design characteristics we propose to measure are important aspects of trial design, then it would indeed be interesting to know that COVID-19 trials are better/worse than those from other indications. If we imagine an extreme example where the control arm were only cancer trials, we think it would still be interesting to know that COVID-19 trials were better or worse: if they are better, this would suggest that we are in fact doing better research than what is generally accepted in a major field of research; if they are worse, this would suggest that investigating COVID-19 is associated with worse trial design than what is generally being done. COVID-19 and cancer are different indications, and the question we are asking is whether investigating one particular indication (COVID-19) is associated with differences in risk of bias compared to investigating other indications. Randomly sampling from those other indications seems a fair way to accurately represent the general standard of research

being conducted outside of the pandemic. We have expanded on this rationale in the protocol:

“The choice of control arm (i.e. non-COVID-19 trial registrations) is important and possibly contentious. For the dataset used for our main analysis (the main dataset) we will randomly sample all non-COVID-19 drug trials, not limiting the indication being studied. We think this is the most relevant comparison because randomly sampling from all trials is a fair way to represent the typical design of trials outside the pandemic, and we want to assess whether the design of COVID-19 trials differs from design of trials outside of the pandemic.”

Having said that, we recognise that others may disagree with this choice of control, so we have considerably expanded what was previously the pneumonia dataset to aim for a sample size equivalent to the main dataset. The dataset now includes additional indications to ensure a sufficient sample size can be met. The main downside to this dataset is that the choice of indications to include is very subjective: there is no clear line between ‘similar’ and ‘not-similar’ indications. For this reason, we have chosen to keep this analysis as a secondary analysis. The protocol text now reads:

“In a second dataset (the indication-matched dataset), our control sample will include only trials which address conditions subjectively comparable to COVID-19. This is to address possible concerns about differences in research design between types of indications (e.g., cancer trials might differ systematically from infectious disease trials) leading to observed differences in the main analysis. We selected respiratory, cardiovascular and infectious disease conditions that have symptoms or treatment approaches similar to COVID-19, then searched these conditions in clinicaltrials.gov and examined the search details for relevant synonyms or similar conditions (selected conditions listed in inclusion criteria). Determining what is a comparable indication is inherently subjective: for example, (non-COVID-19) pneumonia seems an obvious comparator; however, non-COVID-19 pneumonia was the leading cause of death for children under 5 in 2017 (ref: ¹³), whereas the risk to children of COVID-19 is thought to be extremely low¹⁴, so in some senses they are clearly not comparable. Because of this subjectivity, analyses using this dataset will be secondary and included to check the robustness of our findings to changes in the control sample.”

It is not clear to us that differences in the proportion of non-inferiority trials across the two groups will result in biased inferences. We would expect such trials to ideally be designed similarly to other trial types, after conditioning on relevant covariates such as phase.

In terms of outcome types (e.g. survival), we agree that this is an important aspect of trial quality but it is beyond the scope of this study to assess that as an additional outcome that would require significant manual work. We feel that our selected outcomes, although not perfect measures of trial quality, do capture fairly universally important and accepted characteristics of trial design.

Coding blinding dichotomously is inherently problematic, as blinding of outcome assessors and participants are very distinct things. The former is particularly relevant in relation to type of outcome (ie, generally less consequential for hard outcomes, like death), which while the latter hinges on the intervention (eg, in most behavioral interventions, blinding is not possible). I am not sure how considering them interchangeably for a dichotomous rating is informative.

Thank you for this comment. We agree that treating blinding as a dichotomous variable is not ideal. We decided to treat it as a dichotomous variable because we thought that the reporting of blinding would often be ambiguous, so it would be hard to tell exactly who was blinded⁵. However, we explored this in a pilot study and found that who was blinded was often reported. We now propose to collect data on who exactly was blinded, where available. Our main analyses will treat blinding dichotomously because information is still expected to be missing in some cases, but we will also present descriptive data on who was blinded and how this differs by group. We have added further discussion about the limitations of and justification for using blinding in the protocol:

“There are limitations in treating blinding dichotomously as an outcome. For example, the importance of blinding may depend on the subjectivity of the outcome, and who exactly is blinded. There is also uncertainty about the importance of blinding in trials⁴. We have included this outcome because we feel that most assessments of trial quality would include it, but it should be interpreted in the context of these limitations. Where possible we will collect information on who was blinded in the trial and report these data. For the main analysis, we will treat blinding as dichotomous because we expect details of blinding to be missing in some cases (pilot study and refs: ^{5,6}).”

I am also not convinced that the assumptions of no unmeasured confounding stand and that the only relevant confounder to consider is sponsor (for which the authors make a compelling case). For instance, in non-pandemic settings, it is plausible that trials without a comparator are often trials that the sponsor is using for marketing purposes rather than for actually finding if a treatment works. Or, also likely, for interventions where the sponsor is well-aware that effects of an intervention against an adequate comparator would probably be null. For COVID trials, different rationales might apply, for example lack of a comparator might be the consequence, at least early on, of the difficulty of identifying one such comparator. Treatment as usual, already established treatments and so on are valid comparators for non-COVID trials and hence likely to be used. This is not the case for COVID trials. I don't have expertise in causal modelling to allow me to speculate on how examples such as these affect the authors' inference, if they do. But they definitely speak to the possibility that COVID and non-COVID trials might differ in many ways, including the very assumptions under which they are conducted, and these differences are closely intertwined with risk of bias.

Thank you for this comment, which raises important considerations about the potential for unmeasured confounding. We would like to reiterate that we are not arguing that there are no unmeasured confounders, but rather that the interpretation of our models as direct and total effects relies on this assumption. In the section “Sources of residual confounding”, we state: “...it is likely that there will be residual confounding” and “More strictly, when using the term direct effect, we are referring to the association of a trial investigating COVID-19 with the probability of the outcome, conditional on all measured covariates. When using the term total effect, we are referring to the association of a trial investigating COVID-19 with the probability of the outcome, conditional on measured covariates we have identified as confounders: sponsor type and sponsor experience.” In the “Model interpretation” section, we further clarify what we mean by the terms direct and total effects: “they refer to the scenario in which our DAG is correct and complete, and there is no unmeasured confounding”. We also

include sensitivity analysis to explore the strength of association that an unmeasured confounder would need to have to move our point estimate or confidence interval to one (i.e. to no effect) in analysis 8.

Turning to the specific examples raised by the reviewer, we agree that there might be differences in the reasons for including or not including control groups. Regarding the statement that in non-pandemic settings, uncontrolled studies may be for marketing purposes rather than seeing if a treatment actually works, it seems that the same could be said for at least some COVID studies (e.g. an early Remdesivir study⁷). More generally though, even if there is a difference in prevalence of uncontrolled studies specifically for marketing purposes, it is not clear to us that this would change the interpretation of our models – if those studies do not have control groups, they can generally be considered to be methodologically weaker than other studies that do have control groups. Considering the use of treatment as usual/standard of care as a control group, even if the reason for a difference was that COVID did not initially have a standard of care, it is hard to argue that a trial without a control group is ‘better’. We also do not feel that treatment as usual would be an invalid comparator for COVID trials, even early in the pandemic; standard care for patients with e.g. pneumonia or flu could be used, and when little is known about the natural history of the disease it could be argued that the use of controls is even more important. Nevertheless, it might be reasonable to expect that earlier phase trials are more common in the COVID arm compared to the non-COVID arm, and that these might be less likely to be controlled than later phase trials: such differences would be accounted for (at least somewhat) by conditioning on that covariate in the regression models.

We definitely need to know if the COVID trial literature is mostly a big research waste, but I am not convinced that this research proposal is touching on the most relevant questions in this sense. Two important problems of the COVID literature, identified by the surveys of trials planned so far, are overlap (variations of the same research questions in more trials) and the lack of meaningful outcomes (eg, survival, progression to ICU, duration of ICU stay). Though I agree these issues would probably not be labelled risk of bias, they nonetheless would definitely go under methodological quality. For example, a COVID trial without outcome assessor blinding but a hard outcome (eg survival) is arguably more relevant and of better methodological quality than a trial with outcome assessor blinding but a hard to interpret surrogate outcome.

Thank you for this comment. We agree with the reviewer that the other problems highlighted are important, but we also feel that, especially considering the modifications made as a result of this peer review, our study is able to shed light on a relevant question. Investigating outcomes and overlap in research questions is also important, though we feel this needs to be in another research project. At least one paper investigating overlap (multiplicity) has been published¹⁵.

The point about a trial with blinding being more relevant/better methodologically than another with blinding but a surrogate endpoint is well taken, but this in our view highlights an inherent limitation of trying to make comparisons over a large number of studies. There will certainly be cases where the measures (i.e. trial design characteristics) do not fully capture the trial quality, but we are trying to say something in general about the population of trials being

conducted, and studying a large number of trials, and thus omitting some detail on individual trials, is necessary.

As we note in the protocol, because there is differing perception about the importance of individual outcomes, we will analyse them separately rather than combining them into a single measure of 'risk of bias', and, at the suggestion of another reviewers we also now label them as 'design characteristics' rather than 'risk of bias'. For outcomes other than blinding, we feel that the interpretation is much less ambiguous: it is hard to imagine that a non-randomised, uncontrolled or retrospectively registered trial would be considered lower bias than the converse for each particular measure. Blinding is admittedly more controversial, and even its necessity is being challenged by recent meta-analytical work.⁸ However, since it is commonly included in assessments of risk of bias and of trial quality more generally, we feel that it will be of interest to include in this assessment despite the reasonable caveats raised. As detailed above, we have added text further discussing the limitations of using blinding as an outcome.

Reviewer: 4

Comments to the Author(s)

Smith and colleagues propose to investigate differences between reported risks of bias mitigation in clinical trial registrations of COVID-19 intervention trials versus “other” trials on clinicaltrials.gov.

In principle, I think the proposed study is interesting, considered and well-designed but I do have concerns of some of the decisions and assumptions described below. I hope the author's find this of value.

1. I must admit that I am not an expert in trial registries but this study will assess the reporting quality at the time of registration which presents a challenge of the fidelity of the registration to the actual trial design (that you discuss). I suspect this may have been looked at in the past but you can envisage that due to the pressures of the pandemic that registrations are not robustly or comprehensively completed and a likely moderator of the reporting quality will be the completeness of the entry itself.

Thank you for this point. We did not originally think this would be a problem using clinicaltrials.gov data since that is generally well reported. However, as we now expand our analysis to the ICTRP dataset, this is an important consideration as we expect the completeness of the data to be lower. In our analysis, trials that do not state they are blinded, randomised or controlled will be considered open, non-randomised and uncontrolled, so it is possible that incomplete reporting could influence the results. To address this, for those outcomes we will record all instances in which the outcome is assumed based on lack of information, and report values for each group. Sensitivity analysis will be conducted using only those trials with outcomes that we did not have to assume. Analysis 10 now reads:

“We will repeat analyses 4-7 using only those observations for which we did not infer outcome data. This is to check whether decisions to infer outcomes might have differentially affected each group and therefore biased the findings (e.g. if COVID-19 trials tend to include less information as they are more recently registered).”

2. The choice to restrict to only clinicaltrials.gov appears to be driven by the desire to automate the data collection rather than that clinicaltrials.gov is a representative sample of clinical trial registries. A quick look on covid19trialstracker (although the data haven't been updated since 12 August) suggests that almost half the COVID-19 trials are not registered on clinicaltrials.gov. This is an important consideration that could substantially limit the inferences made by the study and a shame given the existence of ICTRP albeit extra manual work.

We are grateful for this comment, which has prompted us to make significant changes to our protocol. As of 12th October 2020, 44% of all COVID-19 trials in the ICTRP database were not registered on clinicaltrials.gov (authors' analysis of data downloaded here: <https://www.who.int/ictcp/en/>). More broadly, outside of COVID-19, there has been an increase in the proportion of studies not registered in clinicaltrials.gov recently.⁹ We therefore agree that limiting the study to focus on clinicaltrials.gov registries may unnecessarily limit the generalizability of our study. We have therefore decided to change our primary analysis to include a representative sample of all trials from ICTRP, and our second dataset will also use ICTRP data. Studying trials outside of clinicaltrials.gov will require significantly more manual work, as some trial registries are challenging or impossible to automate data extraction from. As a result, for the primary analysis, we now limit the sample to a random sample of the size required by our sample size calculation rather than using all available trials which we previously planned to do.

3. Have the authors considered that existing trials that were registered prior to COVID may have adapted their design to add COVID participants? And if this has any implications?

Thank you for this comment. Because we had planned to use a snapshot of the clinicaltrials.gov database from 2019, this would not have been a problem in our original protocol as COVID-19 had not occurred at the time of the snapshot. However, because we are now using ICTRP data, for which a similar snapshot is not available, this is an important consideration. To estimate the prevalence of trials registered prior to COVID-19 adapting their design, we searched clinicaltrials.gov (it is not straightforward to do this using ICTRP, so we use clinicaltrials.gov as a proxy) on 16th October 2020 for COVID-19 trials posted up to 31st December 2019. 25 studies were identified, all of which were posted before the outbreak and appear to have been modified to include COVID-19. Of these, 10 were first posted in 2019 so could plausibly enter into our control sample. This is such a small proportion of trials registered in that year (32,521) that these trials are unlikely to influence our findings.

4. I understand the decision to use a historical sample for the control group as fewer trials are likely to have been registered and I suspect practice is unlikely to have deviated in 1 year but this is an assumption that should probably be checked? The number of Non-COVID intervention registered in 2020 should at least be presented as this would be the obvious comparator group.

Thank you for this comment. The decision to use a historical sample was not based solely on the expectation that the number of clinical trial registrations would have changed, but also that, even when registered, the trials may differ systematically from non-pandemic times. To us, intuitively, the case for a change in the design of registered clinical studies due to the pandemic is much stronger than that for a change in design of clinical studies due to the passage of 1-2 years in time.

As per the reviewer's suggestion, however, we did check the number of trials registered on clinicaltrials.gov in 2019 and 2020, first posted from 1st January to 5th October each year. In 2020 there were 27797 trials, of which 24303 did not study COVID-19, and in 2019 there were 25038. To contextualise this, we looked at the longer term trend by collecting the same data from 2010 to 2018 (see figure). In every year but one, there has been an increase in the number of registered trials. It seems reasonable, therefore, that, in the absence of the pandemic, we would have seen an increase in the number of studies registered; whereas we see a decrease in (non-COVID) studies. This could suggest that the COVID-19 studies are not simply being conducted in addition to normal clinical research, but are instead 'displacing' some studies that would otherwise have been conducted.

Since doing this analysis we have also become aware of a paper suggesting a decrease in new trial starts due to COVID-19 (ref: ¹⁶). This paper is now cited when justifying this decision in the protocol.

We think that this, in combination with the intuitive case, justifies the decision to use the historical sample.

5. I was interested in your choice to look at “at least one form of blinding”? My understanding is that trial entries state whether the design is single or double-blinded. Why is this difference not considered?

Thank you for this comment. The reason for limiting to at least one form of blinding was that we expected information on exactly who was blinded (rather than just single, double, etc) to be poorly reported. In a pilot study we examined this and who was blinded was not reported 19% of the time that blinding occurred. We have therefore decided to conduct our main analysis with blinding treated dichotomously, but also to collect and present information on who was blinded. The relevant part of the paper now reads:

“There are limitations in treating blinding dichotomously as an outcome. For example, the importance of blinding may depend on the subjectivity of the outcome, and who exactly is blinded. There is also uncertainty about the importance of blinding in trials⁴. We have included this outcome because we feel that most assessments of trial quality would include it, but it should be interpreted in the context of these limitations. Where possible we will collect information on who was blinded in the trial and report these data. For the main analysis, we will treat blinding as dichotomous because we expect details of blinding to be missing in some cases (pilot study and refs: ^{5,6}).

6. For the pneumonia sample, I understand that you have chosen a longer period to ensure the sample is sufficiently large but do you have any rules to prioritise the inclusion more recent data once you reach your desired sample size?

Thank you for this suggestion. We did not have rules in place to prioritise more recent data and we agree that we should have. However, the pneumonia dataset has now been changed to be the “indication-matched” dataset, resulting in more trials being available for analysis, and the ability to identify a suitable sample from only 2018 and 2019. We do not think there is a need to prioritise more recent trials in this case.

7. I would encourage the authors to consider a more robust data management plan for manual extraction. It is very easy in Excel to remove, delete or accidentally add data and because of the lack of an audit trail, you may never know this has happened, especially when dealing with large datasets. Systematic review tools are freely available, amenable to this type of research and may provide a useful and robust alternative.

Thank you for this suggestion. We agree that there are some issues with Excel that can be problematic for data management. We did briefly explore the use of systematic review tools such as Rayyan and Covidence, but it seems that they would require significant manipulation to be able to deal with inputs that are not bibliographies (which ours are not). However, we think that Excel is suitable for our purposes, if used appropriately: we intend to use the data validation tools in Excel that are specifically designed to prevent the scenarios outlined above from happening (i.e. removal, deletion or accidental adding of inappropriate data) during data entry, and do extraction in duplicate. Duplicate extraction will be done in separate spreadsheets and then compared using an automated process in R which determines any differences between datasets. As such, unless both extractors happen to delete/modify exactly the same cells in the same way (despite the validation tools), this method will capture any errors introduced during extraction. The data highlighted as

differing will then be examined in the original extraction sheets, thus providing an audit trail of where the errors were introduced. We have added the following text to the paper to explain this: "Data validation tools in Excel will be used to reduce the probability of errors, and extracted datasets will be compared in R."

8. Sensitivity of trial identification – the ICTRP search portal has also created csv/xml outputs for COVID trials. This may be an alternative data source one-step less processed than the

This comment was truncated so we are unsure what it intended to say. However, since we are now using the ICTRP data for our study it seems likely that it is no longer relevant.

References

1. Jones, C. W., Woodford, A. L. & Platts-Mills, T. F. Characteristics of COVID-19 clinical trials registered with ClinicalTrials.gov: cross-sectional analysis. *BMJ Open* **10**, e041276 (2020).
2. Zwierzyna, M., Davies, M., Hingorani, A. D. & Hunter, J. Clinical trial design and dissemination: comprehensive analysis of clinicaltrials.gov and PubMed data since 2005. *BMJ* **361**, (2018).
3. Janiaud, P. *et al.* The worldwide clinical trial research response to the COVID-19 pandemic - the first 100 days. *F1000Res* **9**, 1193 (2020).
4. Moustgaard, H. *et al.* Impact of blinding on estimated treatment effects in randomised clinical trials: meta-epidemiological study. *BMJ* **368**, (2020).
5. Reveiz, L. *et al.* Reporting of methodologic information on trial registries for quality assessment: a study of trial records retrieved from the WHO search portal. *PLoS One* **5**, (2010).
6. Viergever, R. F. & Ghersi, D. Information on blinding in registered records of clinical trials. *Trials* **13**, 210 (2012).
7. Methodological Rigor in COVID-19 Clinical Research: A Systematic Review and Case-Control Analysis | medRxiv. <https://www.medrxiv.org/content/10.1101/2020.07.02.20145102v1>.
8. ClinicalTrials.gov Protocol Registration Data Element Definitions for Interventional and Observational Studies. <https://prsinfo.clinicaltrials.gov/definitions.html>.
9. VanderWeele, T. J. & Ding, P. Sensitivity Analysis in Observational Research: Introducing the E-Value. *Ann. Intern. Med.* **167**, 268–274 (2017).
10. Mathur, M. B., Ding, P., Riddell, C. A. & VanderWeele, T. J. Website and R Package for Computing E-Values. *Epidemiology* **29**, e45–e47 (2018).
11. VanderWeele, T. On a square-root transformation of the odds ratio for a common outcome. *Epidemiology* **28**, e58–e60 (2017).
12. Califf, R. M. Characteristics of Clinical Trials Registered in ClinicalTrials.gov, 2007–2010. *JAMA* **307**, 1838 (2012).
13. Dadonaite, B. & Roser, M. Pneumonia. *Our World in Data* (2018).
14. Mortality Risk of COVID-19 - Statistics and Research. *Our World in Data* <https://ourworldindata.org/mortality-risk-covid>.
15. Kouzy, R. *et al.* Characteristics of the Multiplicity of Randomized Clinical Trials for Coronavirus Disease 2019 Launched During the Pandemic. *JAMA Netw Open* **3**, e2015100 (2020).

16. Xue, J. Z. *et al.* Clinical trial recovery from COVID-19 disruption. *Nature Reviews Drug Discovery* **19**, 662–663 (2020).

Appendix B

Thanks to the reviewer for all of the suggestions. Responses are in italics.

p13. l23 : please provide a link to the histogram that identifies implausible values ;
A link to the file has been added.

- for the models (e.g. p.14-15) should we add a term for residuals ($+ \epsilon$) ?
This has been added.

- Figure 2 appears twice ;
I think this may have been tracked changes; the clean version has only one Figure 2.

- Figure 3 : I would be interested to have both the percentages and the numbers at each time point or at least confidence intervals at each time point for each numbers / the current figure could be a little bit difficult to follow and perhaps a little bit misleading ;
We think that the current figure is consistent with the pre-registered plan: “It is plausible that quality of COVID-19 registrations may have improved or otherwise changed over time. We will examine this by plotting and presenting the prevalence of main outcomes for the trials in the COVID-19 arm of the main dataset over time, grouped into month according to start date. This will be interpreted qualitatively”. The percentages correspond to the proportion which can be read off the y-axis. We are not totally clear on what is difficult to follow or misleading about the figure. For any extra information needed, all data are openly available in the GitHub repository.

- Figure 4 and Table 5 must be merged in the same figure (e.g. like forest plots in meta-analyses that present numerical values) ;
If typesetting allows, these can be printed on the same page immediately adjacent to each other. We experimented with trying to combine them into a single figure but could not make it small enough to fit on a single page, and therefore prefer to keep them separate.

- Table 6 point 5-6 and 7. I don't understand why the interpretation does not make it clear that the positive associations were not robust regarding the differences in control samples;
*This is because the pre-specified interpretation was “To conclude that there was an association between COVID-19 and the outcome¹, we required that the p-value for the analysis of the main dataset was ≤ 0.05 , and that the confidence interval for the direct effect of investigating COVID-19 in the indication-matched dataset (analysis **Error! Reference source not found.**) was consistent with the direction of effect observed in the primary analysis, i.e. that at least part of the confidence interval was in the direction of the effect in the primary analysis, even if the point estimate was not... If the confidence interval for the COVID-19 odds ratio in the main dataset overlapped with 1, we concluded that we did not find evidence of a difference in the outcome prevalence attributable to COVID-19.”*
Part of the confidence interval for point 5 and 6 was in the same direction as the effect in the main analysis and the confidence interval for the main dataset analysis for point 7 overlapped with 1.

¹ The text originally said: “To conclude that there is a difference in the observed proportion” and was corrected to: “to conclude that there was an association between COVID-19 and the outcome”.

Thanks to the reviewer for all of the suggestions. Responses are in italics.

p13. l23 : please provide a link to the histogram that identifies implausible values ;
A link to the file has been added.

- for the models (e.g. p.14-15) should we add a term for residuals ($+ \epsilon$) ?
This has been added.

- Figure 2 appears twice ;
I think this may have been tracked changes; the clean version has only one Figure 2.

- Figure 3 : I would be interested to have both the percentages and the numbers at each time point or at least confidence intervals at each time point for each numbers / the current figure could be a little bit difficult to follow and perhaps a little bit misleading ;
We think that the current figure is consistent with the pre-registered plan: "It is plausible that quality of COVID-19 registrations may have improved or otherwise changed over time. We will examine this by plotting and presenting the prevalence of main outcomes for the trials in the COVID-19 arm of the main dataset over time, grouped into month according to start date. This will be interpreted qualitatively". The percentages correspond to the proportion which can be read off the y-axis. We are not totally clear on what is difficult to follow or misleading about the figure. For any extra information needed, all data are openly available in the GitHub repository.

- Figure 4 and Table 5 must be merged in the same figure (e.g. like forest plots in meta-analyses that present numerical values) ;
If typesetting allows, these can be printed on the same page immediately adjacent to each other. We experimented with trying to combine them into a single figure but could not make it small enough to fit on a single page, and therefore prefer to keep them separate.

- Table 6 point 5-6 and 7. I don't understand why the interpretation does not make it clear that the positive associations were not robust regarding the differences in control samples;
*This is because the pre-specified interpretation was "To conclude that there was an association between COVID-19 and the outcome¹, we required that the p-value for the analysis of the main dataset was ≤ 0.05 , and that the confidence interval for the direct effect of investigating COVID-19 in the indication-matched dataset (analysis **Error! Reference source not found.**) was consistent with the direction of effect observed in the primary analysis, i.e. that at least part of the confidence interval was in the direction of the effect in the primary analysis, even if the point estimate was not... If the confidence interval for the COVID-19 odds ratio in the main dataset overlapped with 1, we concluded that we did not find evidence of a difference in the outcome prevalence attributable to COVID-19."*
Part of the confidence interval for point 5 and 6 was in the same direction as the effect in the main analysis and the confidence interval for the main dataset analysis for point 7 overlapped with 1.

¹ The text originally said: "To conclude that there is a difference in the observed proportion" and was corrected to: "to conclude that there was an association between COVID-19 and the outcome".

- p32 l.7 : "E value interpretation is..." this is in my opinion an interpretation and should be moved to the discussion as it should not appear in the results ;
Agreed, this sentence has been removed.

- First line of the discussion : "our results are inconclusive" is sufficient. I don't think that it is possible to be "inconclusive" and "to point toward a positive effect..." at the same time. I agree that the interpretation is inconclusive and I would therefore delete the discussion about "positive effects".
This has been updated.

- Regarding the comparison with other literature, I would like to see here a discussion of reference 16 that is presented earlier in the manuscript.
This reference is discussed but in the discussion it is reference 58 because the manuscript [58] was published between the stage 1 and stage 2 submission, and the preprint [16] was originally referenced. We have now also referenced number 16 in the discussion to make this clearer.

- In the conclusion, I would not start by stating that these results contradict the previous literature but rather insist on the fact that these results are inconclusive. I really prefer in terms of interpretation the last sentence : "our results are inconclusive but do not support the findings of the related literature". I would also edit the abstract in the same spirit in order to avoid any spin : please delete "the finding contradict. much existing..." and replace with "the findings do not support the existing...".
The conclusion has been updated and now reads: Our findings are inconclusive, but do not support the findings of related literature. The primary analysis generated very strong evidence of a positive effect of COVID-19 on use of control arms and randomisation and weak evidence of a negative effect on blinding. However, secondary and sensitivity analyses challenged the validity of our findings, in some cases reversing the direction of effect. There were also important limitations, including in study design and data quality. The abstract has also been updated as suggested.

- Please also add a few words about limitations in the abstract in order to avoid any spin.
The following was added: "Limitations included some data quality issues, minor deviations from the preregistered plan, and the fact that trial registrations were analysed which may not accurately reflect study design and conduct."

Kudos for this great registered report. I support its publication.
Thanks a lot